# SPOT-JS: SPECTRAL CHEBYSHEV FILTER AND OPTIMAL TRANSPORT FUSION WITH JENSEN-SHANNON ALIGNMENT FOR CROSS-DOMAIN MULTIMODAL DECEPTION DETECTION

## ABSTRACT

Multimodal deception detection is increasingly important for security, justice, and human-AI interaction. However, prevailing systems still depend on contact-based sensing or elaborate handcrafted feature pipelines and exhibit limited generalization beyond their training domains. Typical approaches learn shallow unimodal cues (e.g., surface spatio-temporal patterns) and fuse modalities by simple concatenation or attention; these choices induce sensitivity to positional dependencies and to distribution shift. This work presents SPOT-JS, a frequency-domain framework aimed at cross-domain transfer. It standardizes inputs, improves unimodal representations, and performs fusion with distribution-aware alignment grounded in established theory. Concretely, a Temporal Deception Alignment Module (TDAM) first provides unified preprocessing and audio-visual synchronization to eliminate reliance on specialized facial/vocal features or invasive signals. We then propose a Learnable Chebyshev Spectrum Filter (LCSF) that operates on power spectra to emphasize task-relevant bands and suppress noise by embedding a learnable Chebyshev basis into the spectral transformation. Next, an Optimal Transport-based Cross-Modal Fusion (OTCF) module computes an entropic-regularized transport plan between spectral components of audio and video, enabling fine-grained, bidirectional correspondence and residual fusion in a shared latent space. Fourth, a Jensen-Shannon Guided Alignment (JS-Align) module measures cross-modal posterior similarity via JS divergence and adaptively reweights the fused representation, mitigating sensitivity to positional mismatches and improving stability under shift. Finally, we introduce the Chebyshev Spectrum-guided Knowledge Transfer (CSKT) Module, which leverages spectral filtering to enhance cross-domain facial knowledge transfer. On standard benchmarks (Real Life Trial, DOLOS, and Box of Lies), SPOT-JS surpasses strong unimodal, fusion, and transfer baselines in both intra- and cross-domain settings, with higher F1/ACC/AUC and especially large gains when training on one dataset and testing on another.

## 1 INTRODUCTION

Deception detection is the inference, based on verbal, non-verbal, and/or physiological indicators (e.g. speech content and prosody, facial expression, body motion, gestures, etc.), combined with situational context, of whether a communicator is intentionally misrepresenting the truth. It is a core problem in security, justice and human-AI interaction. Real-world de-

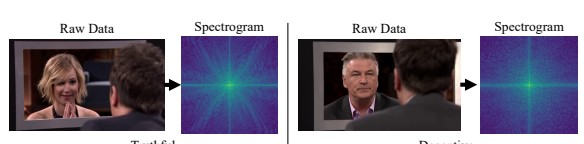

Figure 1: Two examples from the BOL dataset are shown as spectrograms. The spectra display centrally concentrated energy and clear structural patterns.

ceptive behavior has grown more frequent and complex with social change, which makes accurate detection harder. Early practice relied on psychological experts who observed body language, gaze, and facial movement; this approach is informative but requires substantial expertise. Later work

added contact-based measures such as skin conductance, heart rate, and EEG, but these methods require specialized equipment and raise ethical concerns due to their invasiveness.

Traditional data processing pipelines are also too complex. For example, recent work Guo et al. (2024) combines facial frames, OpenFace features (action units and gaze), EmoNet affect metrics (five emotions plus valence-arousal), and both mel spectrograms and raw audio; This choice increases the size of the dataset from roughly 260-2 GB to 20-40 GB.

Automated deception detection using AI and machine learning has become prominent because it scales well, but several obstacles remain: (1) high-accuracy detection still depends on expert knowledge, contact-based physiological signals, or intricate handcrafted features; (2) models transfer poorly to unseen domains due to large scenario differences; (3) unimodal representations are often shallow, emphasizing spatial or temporal cues with limited discriminative power; and (4) multimodal fusion is frequently limited to simple concatenation or attention, which cannot capture fine-grained interactions and is sensitive to positional dependence.

This paper presents **SPOT-JS** (Fig. 2), a frequency-domain framework for cross-domain multimodal deception detection. As shown in Fig. 1, spectrograms exhibit centrally concentrated components and discernible patterns, which motivate our frequency-domain design. First, a Temporal Deception Alignment Module (TDAM) standardizes preprocessing and enforces temporal synchronization between audio and visual streams. Second, a Learnable Chebyshev Spectrum Filter (LCSF) operates on power spectra; a trainable Chebyshev basis highlights task-relevant bands and reduces noise during the spectral transform. Third, an Optimal Transport-based Cross-Modal Fusion (OTCF) module is designed to compute an entropy-regularized transport plan between audio and video spectra to build bidirectional correspondences and perform residual fusion in a shared latent space. Fourth, a Jensen-Shannon Guided Alignment (JS-Align) module adjusts fusion weights according to cross-modal posterior similarity, which improves robustness under complex dependencies. Finally, we introduce the Chebyshev Spectrum-guided Knowledge Transfer (CSKT) Module, which leverages spectral filtering to enhance cross-domain facial knowledge transfer.

This work comprises four technical and an evaluation contributions:

- **TDAM** supplies a unified preprocessing pipeline that removes the reliance on hand-made features or physiological signals and enforces consistent temporal alignment across modalities.

- **LCSF** forms unimodal representations with learnable Chebyshev spectral filtering; it highlights task-relevant frequency bands and suppresses noise in the spectral domain. Building on this, we further develop a **CSKT** module to achieve more effective cross-domain facial knowledge transfer.

- **OTCF** conducts cross-modal fusion in the frequency domain by computing an entropy-regularized transport plan between audio and visual spectra, which yields complementary correspondences and residual fusion in a shared latent space.

- **JS-Align** adjusts fusion weights according to the posterior similarity of the cross-modal and thus reduces the sensitivity to positional dependence.

- Experiments on Real Life Trial, DOLOS, and Box of Lies show gains in both intra-domain and cross-domain tests, with higher F1/ACC/AUC and stable performance under distribution shift.

## 2 RELATED WORK

### 2.1 MULTIMODAL DECEPTION DETECTION

Early research on deception detection examined physiological and behavioral cues under the assumption that fabricating a plausible account taxes cognition. Studies analyzed body language, facial expressions, response patterns, reaction latencies, and pupil dilation (Fitch, 2014; Vrij, 2008). Work on physiological and neural signals used functional magnetic resonance imaging (fMRI) and electroencephalography (EEG) to capture state changes linked to deceptive behavior (Karnati et al., 2021).

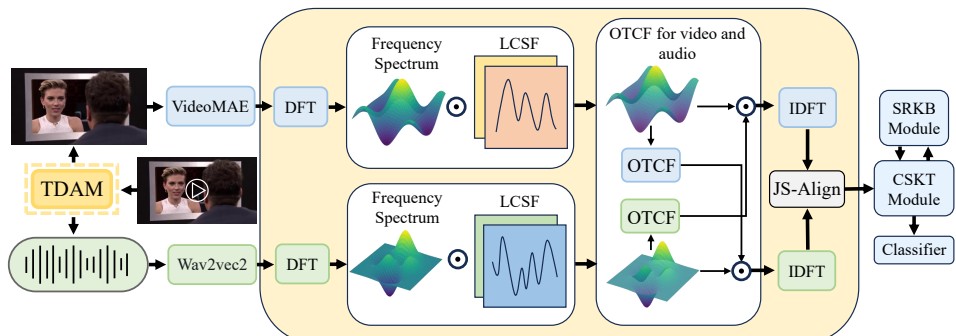

Figure 2: Overall architecture of the proposed method SPOT-JS.

Automated deception detection (ADD) has shifted toward standardized visual toolkits. OpenFace (Amos et al., 2016; Hu et al., 2025b) is now common for video analysis; it uses deep models for facial landmark localization, action unit (AU) prediction (Karimi et al., 2018; Avola et al., 2019), and head pose estimation. Among visual cues, gaze dynamics and facial expression patterns remain central to video-based analysis (Stathopoulos et al., 2020; Mathur & Matarić, 2020; Yildirim et al., 2023; Gallardo Antolín & Montero, 2021).

In speech-based detection, methods extract acoustic markers by examining prosody, pitch contour, and speaking rate. Representative tools include STRAIGHT (Kawahara et al., 2009), voice activity detection (VAD) (Tan & Lindberg, 2010), and openSMILE (Eyben et al., 2010); Mel-frequency cepstral coefficients (MFCCs) (Mermelstein, 1976; Davis & Mermelstein, 1980) are widely used features.

For multimodal fusion, Guo et al. (2023) introduced a dataset and framework leveraging audio-visual complementarity for deception detection. Li et al. (2024) achieved synchronized integration by aligning temporal embeddings, while Ji et al. (2025) advanced knowledge transfer with a hierarchical optimal transport approach using large-scale facial expression priors.

## 2.2 FREQUENCY DOMAIN LEARNING

Recent studies move feature learning into the frequency domain. Xu et al. (2020) reported a learning-based filter that removes trivial frequency components and improves image classification. In text classification, Lee-Thorp et al. (2022) applied the Fourier transform for token mixing. In time-series forecasting, Yang & Hong (2022) introduced Bilinear Time-Spectral Fusion, which models time-frequency pairs and uses spectral-to-temporal and temporal-to-spectral aggregation to update representations. In the field of rumor detection, Lao et al. (2024) proposed a novel dual contrastive learning-based spectral representation and fusion network.

## 3 PROBLEM DEFINITION

Multimodal deception detection is formulated as a binary classification task with modalities $t \in \{a, v\}$ for audio ($a$) and video ($v$). Given a multimodal deception dataset $\mathcal{D} = \{\mathcal{X}, \mathcal{Y}\}$, each instance is denoted as a two-tuple $(x, y) \in \mathcal{D}$, where $x \in \mathcal{X}$ and $y \in \mathcal{Y}$. Specifically, $x$ can be represented as $x = \{x^a, x^v\}$. The label space is defined as $y = \{0, 1\}$, where $y = 1$ denotes a deceptive sample and $y = 0$ a truthful one. The goal of this work is to learn a decision function $f : \mathcal{X} \to \mathcal{Y}$ that effectively utilizes multimodal features to predict the deception label $\hat{y} \in \{0, 1\}$.

# 4 METHOD

## 4.1 TEMPORAL DECEPTION ALIGNMENT MODULE (TDAM)

The Temporal Deception Alignment Module (TDAM) preprocesses raw video and audio to give temporal consistency and a common feature format before downstream analysis. TDAM has two components.

**Video preprocessing.** For a raw video sequence $x^v$ of duration $T$, TDAM applies uniform temporal sampling to obtain $N$ key frames:

$$f_i = x^v \left( \tau_0 + i \cdot \frac{T}{N} \right) \tag{1}$$

where $x^v(\tau)$ denotes the frame at timestamp $\tau$, $\tau_0$ is the start time, and $\tau \in [0, T]$. Each sampled frame $f_i$ is converted from BGR to RGB and mapped to a PIL image via $\phi(\cdot)$. A proposed transformation $\mathcal{T}(\cdot)$ then standardizes the frame sequence, producing a normalized video tensor:

$$\hat{x}^v = \mathcal{T}\left(\{\phi(f_i)\}_{i=1}^N\right), \qquad \hat{x}^v \in \mathbb{R}^{B \times 3 \times H \times W}. \tag{2}$$

**Audio preprocessing.** From the same video, TDAM extracts the audio track and resamples it to align with the visual stream:

$$\hat{x}^a = \mathcal{R}\left(\mathcal{A}(x^v), f_s'\right), \qquad f_s' = \frac{N}{T} \cdot f_s \tag{3}$$

where $\mathcal{A}(\cdot)$ extracts the raw audio, $\mathcal{R}(\cdot)$ denotes resampling, and $f_s$ and $f_s'$ represent the original and adjusted sampling rates, respectively.

The two-stream preprocessing yields temporally coherent modalities and a standardized input for subsequent training and inference.

## 4.2 FEATURE ENCODING

After TDAM, the normalized video tensor $\hat{x}^v$ and the resampled audio signal $\hat{x}^a$ are passed to pretrained backbones for feature encoding. For the visual stream, we employ VideoMAE-Base (Tong et al., 2022), which encodes the sampled video frames into spatio-temporal representations:

$$\mathbf{x}^v = \text{VideoMAE}(\hat{x}^v) \tag{4}$$

For the audio stream, we adopt Wav2Vec2-Base (W2V2) (Baevski et al., 2020), which transforms the waveform into a sequence of latent speech features:

$$\mathbf{x}^a = \text{W2V2}(\hat{x}^a) \tag{5}$$

Accordingly, let $\mathbf{x}^t$ denote the encoded sequence for modality $t \in \{a, v\}$. These sequences are then fed into the subsequent spectral modules (LCSF, OTCF, JS-Align and CSKT).

## 4.3 SPECTRUM REPRESENTATION

Spatial features are transformed into spectrum features via the Discrete Fourier transform (DFT). The spectra of the video and audio embeddings are given by

$$\mathbf{X}^t[k] = \mathcal{F}_{seq}(\mathbf{x}^t[i]) = \sum_{i=0}^{N-1} \mathbf{x}^t[i] e^{-j(2\pi/N)ki} \tag{6}$$

where $\mathbf{X}^t \in \mathbb{C}^{B \times N \times D}$ for $t \in \{a, v\}$ is a complex-valued tensor, $\mathbf{X}^t[k]$ denotes the spectrum of $\mathbf{x}^t[i]$ at frequency $2\pi k/N$, $\mathcal{F}_{seq}(\cdot)$ is the 1-Dimension (1D) DFT along the sequence dimension, and $j$ is the imaginary unit.

### 4.4 LEARNABLE CHEBYSHEV SPECTRUM FILTER (LCSF) MODULE

We present a new unimodal spectral filter module derived from the Chebyshev formulation, referred to as Learnable Chebyshev Spectrum Filter (LCSF). For a spectrum $\mathbf{X}^t$ with $t \in \{a, v\}$, the power spectrum $|\mathbf{X}^t|^2$ is first computed to focus on primary intra-modal patterns. A learnable Chebyshev coefficient set $\mathbf{C}^t = [C_1^t, C_2^t, \ldots, C_k^t]$ is combined with a filter bank $\mathbf{K}^t = [\mathbf{k}_1^t, \mathbf{k}_2^t, \ldots, \mathbf{k}_k^t]$ to form the transformed representation:

$$\hat{\mathbf{X}}^t = \sum_{i=1}^{k} |\mathbf{X}^t|^2 \odot \mathbf{k}_i^t \, C_i^t, \tag{7}$$

where $\odot$ indicates element-wise multiplication. The coefficients are defined from the Chebyshev formulation as:

$$C_i^t = \cos\big((2i-1)\,\theta_{\text{base}}\big), \quad \theta_{\text{base}} = \text{softplus}(\alpha) \cdot \tfrac{\pi}{2k}, \tag{8}$$

with $\alpha$ as a trainable parameter. This design embeds the Chebyshev structure into the spectral transformation, allowing the model to adaptively adjust the spectral basis and capture more discriminative unimodal information in the frequency domain.

### 4.5 OPTIMAL TRANSPORT-BASED CROSS-MODAL FUSION (OTCF) MODULE

**Optimal Transport Theory.** Optimal transport (OT) (Peyré & Cuturi, 2019) seeks a minimal-cost map from one probability distribution to another. Let $p = \sum_{i=1}^{n} a_i \delta_{\mathbf{X}_{A_i}}$ and $q = \sum_{j=1}^{m} b_j \delta_{\mathbf{X}_{B_j}}$ be $n$ and $m$ dimensional discrete probability distributions for two finite sets $\mathbf{X}_A = \{\mathbf{X}_{A_i}\}_{i=1}^{n}, \mathbf{X}_B = \{\mathbf{X}_{B_j}\}_{j=1}^{m}$ respectively, where $\boldsymbol{a} \in \Delta_n$ and $\boldsymbol{b} \in \Delta_m$, $\Delta_n$ and $\Delta_m$ are the probability simplex of $\mathbb{R}^n$ and $\mathbb{R}^m$, and $\delta_{\mathbf{X}_*}$ refers to a point mass located at coordinate $\mathbf{X}_* \in \mathbb{R}^d$. Denoting $\mathbf{M} \in \mathbb{R}_+^{n \times m}$ as the cost matrix with $\mathbf{M}_{i,j} = \mathcal{M}(\mathbf{X}_{A_i}, \mathbf{X}_{B_j})$, which means the cost to transport one unit of mass between elements of the sets. Then, the transport plan matrix $\mathbf{T}$ is obtained by solving:

$$\text{OT}(p, q) = \min_{\mathbf{T} \in \Pi(p,q)} \langle \mathbf{T}, \mathbf{M} \rangle_{\text{F}} \tag{9}$$

where $\langle \cdot, \cdot \rangle_{\text{F}}$ is the Frobenius dot-product. The constraint $\Pi(p, q) := \{\mathbf{T} \in \mathbb{R}_+^{n \times m} \mid \sum_{i=1}^{n} \mathbf{T}_{i,j} = b_j, \sum_{j=1}^{m} \mathbf{T}_{i,j} = a_i\}$ enforces $\mathbf{T}$ to have $p, q$ as its marginals. It should be noted that $\mathbf{T}$ can be interpreted as the probabilistic correspondence between the elements of $p$ and $q$. If the transport cost $\mathbf{M}_{i,j}$ between $\mathbf{X}_{A_i}$ and $\mathbf{X}_{B_j}$ is high, then a low correlation $\mathbf{T}_{i,j}$ should be obtained. Eq. (9) is a linear assignment problem, which is expensive to solve. Fortunately, an entropy-regularized OT has been developed as follows:

$$\text{OT}(p, q) = \min_{\mathbf{T} \in \Pi(p,q)} \langle \mathbf{T}, \mathbf{M} \rangle_{\text{F}} - \epsilon \mathcal{H}(\mathbf{T}), \tag{10}$$

where $\mathcal{H}(\mathbf{T}) = -\sum_{i,j} \mathbf{T}_{i,j} \log \mathbf{T}_{i,j}$ is the entropic regularization. Eq. (10) can be solved efficiently by the log-domain Sinkhorn algorithm Cuturi (2013).

**OTCF Module.** For clarity, audio features are treated as target and video features as source; the reverse direction is symmetric. OTCF maps $\hat{\mathbf{X}}^v$ and $\hat{\mathbf{X}}^a$ to a shared latent space via two linear projections:

$$\tilde{\mathbf{X}}^v = \hat{\mathbf{X}}^v \mathbf{W}_s, \qquad \tilde{\mathbf{X}}^a = \hat{\mathbf{X}}^a \mathbf{W}_t, \qquad \mathbf{W}_s, \mathbf{W}_t \in \mathbb{R}^{D \times D}, \qquad \tilde{\mathbf{X}}^v, \tilde{\mathbf{X}}^a \in \mathbb{R}^{B \times N \times D} \tag{11}$$

The ground cost uses cosine distance between the projected sequences:

$$\mathbf{M} = 1 - \cos(\tilde{\mathbf{X}}^v, \tilde{\mathbf{X}}^a) \tag{12}$$

computed with a cosine-similarity kernel.

For each batch element, the entropic OT in Eq. (10) is solved by log-stabilized Sinkhorn iterations. Solving Eq. (10) yields the transport plan $\mathbf{T}$, which is then used in subsequent steps. The plan $\mathbf{T}$ provides a soft, mass-conserving alignment from source to target. OTCF forms a fused source representation by transporting and aggregating target features and then adding a residual term:

$$\mathbf{z}^v = \mathbf{T}\,\tilde{\mathbf{X}}^a + \tilde{\mathbf{X}}^v \tag{13}$$

In the full model, OTCF is applied in both directions (visual→audio and audio→visual) with the same formulation and independent projection parameters.

Finally, we employ inverse discrete Fourier transform (IDFT, $\mathcal{F}_{seq}^{-1}$) to convert the spectral representations of audio and video back into the spatial domain:

$$\mathbf{Z}^t \leftarrow \mathcal{F}_{seq}^{-1}(\mathbf{z}^t), t \in \{a, v\} \tag{14}$$

### 4.6 JENSEN-SHANNON GUIDED ALIGNMENT (JS-ALIGN) MODULE

With audio and visual representations refined by optimal transport-based cross-modal fusion (OTCF), the distributions of the two modalities are compared using the JensenShannon (JS) divergence to guide the subsequent fusion step. The distributional divergence is

$$\mathrm{J} = \mathrm{JS}(\mathbf{Z}^v\|\mathbf{Z}^a) = \tfrac{1}{2}\,\mathrm{KL}(\mathbf{Z}^v \parallel M) + \tfrac{1}{2}\,\mathrm{KL}(\mathbf{Z}^a \parallel M) \tag{15}$$

where $\mathrm{JS}(\cdot)$ denotes the JS divergence, $\mathrm{KL}(\cdot)$ represents the KL divergence, $M = \tfrac{1}{2}(\mathbf{Z}^v + \mathbf{Z}^a)$ is the mean distribution, and J serves as a similarity score between modalities.

The multimodal representation is then obtained as:

$$\mathbf{f} = (1 - \mathrm{J})(\mathbf{W}^a\mathbf{Z}^a + \mathbf{W}^v\mathbf{Z}^v) + \mathrm{J}\,\mathbf{Z}^a + \mathrm{J}\,\mathbf{Z}^v \tag{16}$$

with $\mathbf{W}^a$ and $\mathbf{W}^v$ trainable parameters. Here J is computed from the JS divergence, so the fusion weights reflect cross-modal similarity. In unimodal settings, only Eqs. (6) to (8) and (14) are used; the complete procedure applies in multimodal settings.

### 4.7 CHEBYSHEV SPECTRUM-GUIDED KNOWLEDGE TRANSFER (CSKT) MODULE

The fused or single-modality features are passed to the Chebyshev Spectrum-guided Knowledge Transfer (CSKT) module. CSKT extends Hierarchical Optimal Transport Knowledge Transfer (H-OTKT) (Ji et al., 2025) with the Learnable Chebyshev Spectrum Filter (LCSF) and uses the Sample-specific Re-weighting Knowledge Bank (SRKB) from Ji et al. (2025) to transfer facial knowledge $\mathbf{X}^s$ distilled from DFEW (Jiang et al., 2020). Further details appear in Appendix Section C.

$$\mathbf{X}^{fused} = \mathbf{CSKT}(\mathbf{f}, \mathbf{X^s}) \tag{17}$$

## 5 CLASSIFICATION

The final classification layer contains one MLP with softmax, which takes $\mathbf{X}^{fused}$ as input and outputs the predicted label $\hat{\boldsymbol{y}} \in \mathbb{R}^{n \times L^t}$:

$$\hat{\boldsymbol{y}} = \mathcal{F}_3(\mathbf{X}^{fused}) \tag{18}$$

Here, $\mathcal{F}_3$ is the MLP classifier. With ground truth label $\boldsymbol{y} = [y_1, \ldots, y_n]$, the classification loss function is formulated as:

$$\mathcal{L}_{ce}(\boldsymbol{y}, \hat{\boldsymbol{y}}) = -\mathbb{E}_{\boldsymbol{y}}[\log \hat{\boldsymbol{y}}] \tag{19}$$

where $\mathbb{E}$ is expectation. To reduce the difference between distribution spaces from source and target domain in H-OTKT, and further improve the final prediction, another loss function is defined based on the Sinkhorn divergence (Feydy et al., 2019) to obtain the space discrepancy between class average of $\mathbf{X}^s$ and $\mathbf{f}'$ (Nguyen & Luu, 2022):

$$\mathcal{L}_{ot}(\mathbf{f}', \mathbf{X}^s) = ds_{\mathrm{OT}}(\mathcal{P}, \mathcal{Q}) - \tfrac{1}{2}ds_{\mathrm{OT}}(\mathcal{P}, \mathcal{P}) - \tfrac{1}{2}ds_{\mathrm{OT}}(\mathcal{Q}, \mathcal{Q}) \tag{20}$$

where $ds_{\mathrm{OT}}(\cdot, \cdot)$ is the total OT cost between two distributions solved by the regular OT (Eq. (9)) with cosine similarity as cost function. Then the total loss function is formulated as:

$$\mathcal{L} = \mathcal{L}_{ce} + \eta\mathcal{L}_{ot} \tag{21}$$

In Eq. (21), the $\mathcal{L}_{ce}$ term optimizes the whole network to improve the classification performance while the $\mathcal{L}_{ot}$ term is used for reducing the discrepance between the source feature space and the target feature space.

Table 1: Results in Real Life Trial (RLT) dataset and DOLOS dataset

(a) Results with visual modality.

| Target | RLT | | | DOLOS | | |
|---|---|---|---|---|---|---|
| Method | F1 score | ACC | AUC | F1 score | ACC | AUC |
| OpenFace + SVM | 0.2253 | 0.5293 | 0.5571 | 0.6975 | 0.5355 | 0.5430 |
| OpenFace + Decision Tree | 0.5553 | 0.5303 | 0.5303 | 0.5358 | 0.5058 | 0.5058 |
| OpenFace + Random Forest | 0.6033 | 0.6033 | 0.5997 | 0.6175 | 0.5367 | 0.5466 |
| OpenFace + AdaBoost | 0.5199 | 0.5303 | 0.5766 | 0.5536 | 0.5057 | 0.5035 |
| AU + SVM | 0.4562 | 0.5043 | 0.4670 | 0.6813 | 0.5276 | 0.5242 |
| AU + Decision Tree | 0.4466 | 0.4643 | 0.4643 | 0.5453 | 0.5173 | 0.5173 |
| AU + Random Forest | 0.5534 | 0.5463 | 0.5330 | 0.5808 | 0.5045 | 0.5157 |
| AU + AdaBoost | 0.5130 | 0.4877 | 0.4835 | 0.5295 | 0.4876 | 0.4735 |
| OpenFace + LSTM | 0.5241 | 0.5623 | 0.5952 | 0.5928 | 0.5628 | 0.5854 |
| AU + LSTM | 0.4888 | 0.6197 | 0.6760 | 0.6343 | 0.5646 | 0.5868 |
| ResNet18 + LSTM | 0.4996 | 0.6117 | 0.6387 | 0.6415 | 0.5972 | 0.5668 |
| PECL(only visual) | 0.5880 | 0.6528 | 0.6734 | 0.7010 | 0.6387 | 0.6770 |
| FreeLunch | 0.7663 | 0.8173 | 0.8712 | 0.6961 | 0.6228 | 0.6459 |
| ADC | 0.7793 | 0.8173 | 0.8677 | 0.6938 | 0.6716 | 0.7206 |
| Cr-KD-NCD | 0.7805 | 0.7200 | 0.6928 | 0.7056 | 0.6091 | 0.6013 |
| AFFAKT | 0.8760 | 0.8670 | 0.8789 | 0.7102 | 0.6764 | 0.7212 |
| SPOT-JS(Ours) | 0.9600 | 0.9600 | 0.9948 | 0.9643 | 0.9649 | 0.9692 |
| | +8.40% | +9.30% | +11.59% | +25.41% | +28.85% | +24.80% |

(b) Results with audio modality.

| Target | RLT | | | DOLOS | | |
|---|---|---|---|---|---|---|
| Method | F1 | ACC | AUC | F1 | ACC | AUC |
| MFCC + MLP | 0.5226 | 0.6367 | 0.7030 | 0.5963 | 0.5810 | 0.6134 |
| OpenSMILE + MLP | 0.6885 | 0.6597 | 0.5926 | 0.6867 | 0.5537 | 0.5325 |
| W2V2 + MLP | 0.6117 | 0.6780 | 0.6106 | 0.4383 | 0.5421 | 0.5369 |
| PECL(only audio) | 0.7121 | 0.7100 | 0.6962 | 0.6777 | 0.6119 | 0.6281 |
| FreeLunch | 0.6432 | 0.6850 | 0.6944 | 0.6589 | 0.5991 | 0.6196 |
| ADC | 0.6402 | 0.6767 | 0.6858 | 0.6196 | 0.6058 | 0.6040 |
| AFFAKT | 0.7316 | 0.7440 | 0.7396 | 0.6982 | 0.6198 | 0.6391 |
| SPOT-JS(Ours) | 0.8571 | 0.8333 | 0.8715 | 0.7463 | 0.7143 | 0.7659 |
| | +12.55% | +8.93% | +13.19% | +4.81% | +9.45% | +12.68% |

(c) Results with fused modalities.

| Target | RLT | | | DOLOS | | |
|---|---|---|---|---|---|---|
| Method | F1 | ACC | AUC | F1 | ACC | AUC |
| OpenFace ⊕ OpenSMILE | 0.6895 | 0.6781 | 0.6212 | 0.6124 | 0.5986 | 0.5863 |
| ResNet18 ⊕ OpenSMILE | 0.6283 | 0.6853 | 0.6598 | 0.5863 | 0.6152 | 0.6485 |
| PECL | 0.7102 | 0.6939 | 0.7424 | 0.7084 | 0.6597 | 0.6353 |
| FreeLunch | 0.7695 | 0.8093 | 0.8547 | 0.6807 | 0.6289 | 0.6574 |
| ADC | 0.7493 | 0.8093 | 0.8446 | 0.6997 | 0.6746 | 0.7307 |
| AFFAKT | 0.8412 | 0.8427 | 0.8563 | 0.7149 | 0.6810 | 0.7289 |
| SPOT-JS(Ours) | 0.9630 | 0.9600 | 0.9679 | 0.9474 | 0.9474 | 0.9846 |
| | +12.18% | +11.73% | +11.16% | +23.25% | +26.64% | +25.57% |

Table 2: Results in Box of lies(BOL) dataset.

(a) Results with visual modalities.

| Target | Box of lies | | |
|---|---|---|---|
| Method | F1 | ACC | AUC |
| CMFL | 0.5584 | 0.4403 | 0.4907 |
| SE-Concat | 0.5606 | 0.5678 | 0.5657 |
| Prompt | 0.6785 | 0.5451 | 0.6143 |
| PECL | 0.6832 | 0.5705 | 0.6476 |
| AVA+CUFMCL | 0.6953 | 0.5947 | 0.6743 |
| SPOT-JS(Ours) | 0.9333 | 0.9111 | 0.9551 |
| | +23.80% | +31.64% | +28.08% |

(b) Results with audio modalities.

| Target | Box of lies | | |
|---|---|---|---|
| Method | F1 | ACC | AUC |
| CMFL | 0.5812 | 0.5308 | 0.5406 |
| SE-Concat | 0.5694 | 0.5530 | 0.5638 |
| Prompt | 0.6684 | 0.5673 | 0.5799 |
| PECL | 0.6726 | 0.5828 | 0.6122 |
| AVA+CUFMCL | 0.6972 | 0.5987 | 0.6456 |
| SPOT-JS(Ours) | 0.9310 | 0.9111 | 0.9131 |
| | +23.38% | +31.24% | +26.75% |

(c) Results with fused modalities.

| Target | Box of lies | | |
|---|---|---|---|
| Method | F1 | ACC | AUC |
| CMFL | 0.6568 | 0.5350 | 0.5635 |
| SE-Concat | 0.6721 | 0.5919 | 0.6109 |
| Prompt | 0.6891 | 0.5954 | 0.6256 |
| PECL | 0.6723 | 0.6078 | 0.6433 |
| AVA+CUFMCL | 0.6920 | 0.6256 | 0.6667 |
| SPOT-JS(Ours) | 0.9153 | 0.8889 | 0.9393 |
| | +22.33% | +26.33% | +27.26% |

# 6 EXPERIMENTS

## 6.1 DATASETS

**Datasets.** To validate the effectiveness, generalizability, and robustness of SPOT-JS, evaluation is performed on three benchmark datasets: (1) Real Life Trial (Pérez Rosas et al., 2015), (2) Box of Lies (BOL)(Soldner et al., 2019), (3) DOLOS (Guo et al., 2023). For detailed information about the dataset, please refer to Appendix Section G.1.

## 6.2 COMPARISON METHODS

Machine learning methods typically employ visual features (OpenFace, gaze and action units) and acoustic features (MFCC) for deception detection. Standard classifiers such as SVM and Decision Tree process the visual features, while MLP handles the acoustic features (Mathur & Matarić, 2020; Avola et al., 2019; Yang et al., 2021a).

Deep learning approaches include several architectures: KNN (Chebbi & Jebara, 2023), FFCSN (Ding et al., 2019b), ResNet18+LSTM (Karnati et al., 2022; Ding et al., 2019a; Guo et al., 2023), W2V2+MLP (Guo et al., 2023; Karnati et al., 2022; Krishnamurthy et al., 2018), ResNet18⊕OpenSMILE (Krishnamurthy et al., 2018; Guo et al., 2023), CLBF (Camara et al., 2024), and PECL (Guo et al., 2023).

Fusion methods include: Concat, SE-Concat (Hu et al., 2018), CMFL (George & Marcel, 2021), Prompt (Jia et al., 2022), and AVA (Li et al., 2024).

Transfer learning methods include: FreeLunch (Yang et al., 2021b), ADC (Guo et al., 2022), PECL (Guo et al., 2023), Cr-KD-NCD (Gu et al., 2023), and AFFAKT (Ji et al., 2025).

## 6.3 COMPARISON RESULTS

Experiments were conducted on three datasets-Real Life Trial (RLT), DOLOS, and Box of Lies (BOL) under visual, audio, and fused modalities. Performance was measured by F1, ACC, and AUC with 5-fold cross-validation. Our method achieved significant improvements: on the RLT dataset, ACC improved by 8.93%–11.73% and on the DOLOS dataset, ACC increased by 9.45%–28.85% in

Tables 1a to 1c; and on the Box of Lies dataset, ACC saw a gain of 26.33%–31.64% in Tables 2a to 2c.The same tables show consistent increases in F1 and AUC across all three datasets.

Cross-domain evaluations report ACC; the remaining metrics appear in Appendix Section E. As shown in Tables 3a to 3c, the results exceed strong baselines across modalities. For the visual stream, training on RLT (R) and testing on BOL (B) gives a 25.76% gain over prior methods. For the audio stream under the same setup, the gain is 23.76%. For the fused modality, training on BOL and testing on RLT yields 35.61%. Similar margins appear in other transfer directions. These results indicate robust cross-domain performance of the proposed approach.

Table 3: Cross-domain results on RLT (R), BOL (B), and DOLOS (D) using ACC. Detailed results are provided in Appendix Section E. Abbreviations: Ff, face frames; Mel, Mel spectrogram.

(a) Results with visual modality.

| Method | R to D | D to B | R to B | D to R | B to R | B to D |
|---|---|---|---|---|---|---|
| AU+LSTM | 0.4992 | 0.4997 | 0.5886 | 0.5233 | 0.5489 | 0.5046 |
| Gaze+MLP | 0.4998 | 0.5011 | 0.5998 | 0.5308 | 0.5519 | 0.5008 |
| AU+Gaze+MLP | 0.5125 | 0.5137 | 0.6535 | 0.5479 | 0.5563 | 0.5153 |
| Affect+MLP | 0.5132 | 0.5108 | 0.5842 | 0.5458 | 0.5587 | 0.5226 |
| AU+Gaze+Affect+MLP | 0.5238 | 0.5025 | 0.5941 | 0.5541 | 0.5633 | 0.5289 |
| Ff+ResNet18 | 0.5275 | 0.5133 | 0.6139 | 0.5452 | 0.5645 | 0.5238 |
| Ff+ResNet18+GRU | 0.5254 | 0.5236 | 0.6337 | 0.5480 | 0.5688 | 0.5278 |
| Ff+ResNet18+KNN | 0.5337 | 0.5148 | 0.6315 | 0.5536 | 0.5609 | 0.5308 |
| Ff+ResNet18+SVM | 0.5415 | 0.5028 | 0.6328 | 0.5636 | 0.5709 | 0.5382 |
| Ff+FFCSN | 0.5387 | 0.5319 | 0.6288 | 0.5685 | 0.5682 | 0.5408 |
| CLBF | 0.5283 | 0.5136 | 0.4409 | 0.5458 | 0.5864 | 0.5538 |
| PECL | 0.5516 | 0.5307 | 0.6399 | 0.5532 | 0.5277 | 0.5509 |
| SPOT-JS(Ours) | 0.7143 | 0.7778 | 0.9111 | 0.7200 | 0.7083 | 0.6140 |
| | +16.27% | +24.59% | +25.76% | +15.15% | +12.19% | +6.02% |

(b) Results with audio modality.

| Method | R to D | D to B | R to B | D to R | B to R | B to D |
|---|---|---|---|---|---|---|
| Acoustic + Prosodic+MLP | 0.4558 | 0.5022 | 0.5218 | 0.5038 | 0.5126 | 0.4947 |
| Mel+ResNet18 | 0.5001 | 0.5386 | 0.5347 | 0.5343 | 0.5256 | 0.4907 |
| Mel+ResNet18+KNN | 0.4882 | 0.5238 | 0.5402 | 0.5317 | 0.5346 | 0.4899 |
| Mel+ResNet18+SVM | 0.4905 | 0.5398 | 0.5402 | 0.5425 | 0.5391 | 0.5006 |
| Waveform+Wave2Vec | 0.5021 | 0.5365 | 0.4851 | 0.5355 | 0.5309 | 0.5087 |
| PECL | 0.5197 | 0.5368 | 0.5392 | 0.5915 | 0.5447 | 0.5125 |
| SPOT-JS(Ours) | 0.6429 | 0.6889 | 0.7778 | 0.6400 | 0.7083 | 0.6316 |
| | +12.32% | +14.91% | +23.76% | +4.85% | +16.36% | +11.91% |

(c) Results with fused modality.

| Method | R to D | D to B | R to B | D to R | B to R | B to D |
|---|---|---|---|---|---|---|
| Average | 0.5385 | 0.5822 | 0.5842 | 0.5338 | 0.4907 | 0.5089 |
| Concat | 0.5393 | 0.5832 | 0.5842 | 0.5625 | 0.4982 | 0.5103 |
| SE-Concat | 0.5339 | 0.5945 | 0.6040 | 0.5695 | 0.5069 | 0.5169 |
| Cross-Atten | 0.5411 | 0.5941 | 0.6139 | 0.5733 | 0.5166 | 0.5237 |
| MLP-Mixer | 0.5497 | 0.6042 | 0.5743 | 0.5754 | 0.5283 | 0.5369 |
| PECL | 0.5601 | 0.6136 | 0.5967 | 0.5617 | 0.5319 | 0.5427 |
| Atten-Mixer | 0.5635 | 0.6337 | 0.6040 | 0.5877 | 0.5256 | 0.5433 |
| SPOT-JS(Ours) | 0.7018 | 0.7333 | 0.7556 | 0.7500 | 0.8880 | 0.6491 |
| | +13.83% | +9.96% | +14.17% | +16.23% | +35.61% | +10.08% |

## 6.4 Implementation Details

Experiments ran on an NVIDIA RTX 4090 GPU (24GB VRAM). For video and fused modalities across DOLOS, RLT, and BOL, the batch size was 16; for the audio modality, a batch size of 64 was used due to lower memory demand. The learning rate was fixed at $1 \times 10^{-5}$ throughout. Additional implementation notes appear in Appendix Section G.2.

## 6.5 Ablation Studies

Our ablation study systematically validates the effectiveness of the proposed spectral representation and fusion method. As shown in the table, we design three experimental scenarios for comprehensive evaluation.

Case A. As a purely baseline setting, our first ablation excludes all key innovations, retaining only the standard H-OTKT and SRKB modules. As shown in Table 4, the baseline version exhibits a pronounced performance degradation compared to the full model.

Case B. To validate the effectiveness of TDAM, we introduce this module on top of Method A. The experimental results show that all three modalities achieve significant improvements. These findings demonstrate that: (1) preserving inter-frame continuity in video sequences is essential; and (2) minimizing noise while maintaining temporal consistency plays a crucial role in enhancing the model's robustness and generalization ability.

Case C. Building upon Case B, we incorporate the LCSF module. As shown in Table 4, all three modalities achieve substantial improvements over the baseline model. This demonstrates that LCSF effectively highlights task-relevant frequency bands while suppressing noise through learnable Chebyshev spectral filtering.

Case D. Building upon Method C, we replace H-OTKT with our modified CSKT. The experimental results reveal significant improvements across all three modalities, indicating that the previous direct knowledge transfer approach was rather coarse. With the introduction of our CSKT module, facial knowledge is transferred more effectively, which further substantiates the effectiveness of the LCSF module.

Case E. Building on Case D, we integrate the OTCF module into the multimodal setting. The experimental results demonstrate that, compared with the mixed-modal scheme in Case D, adopting the OTCF approach leads to improved model performance.

Case F. Building on Case D, we integrate the JS-Align module into the multimodal setting. The experimental results show that, compared with the mixed-modal scheme in Case D, adopting the JS-Align method consistently enhances model performance.

Due to space limitations, additional ablation results are provided in the Appendix Section F.

Table 4: Ablation studies results. ① TDAM module, ② LCSF module, ③ CSKT module, ④ OTCF module, ⑤ JS-Align module.

| Case | Target Method ① | ② | ③ | ④ | ⑤ | modality | RLT F1 | ACC | AUC | DOLOS F1 | ACC | AUC | BOL F1 | ACC | AUC |
|---|---|---|---|---|---|---|---|---|---|---|---|---|---|---|---|
| A | ✗ | ✗ | ✗ | ✗ | ✗ | Visual | 0.8760 | 0.8670 | 0.8789 | 0.7054 | 0.6764 | 0.7212 | 0.7302 | 0.6889 | 0.7377 |
| | | | | | | Audio | 0.7267 | 0.7270 | 0.7218 | 0.6822 | 0.6198 | 0.6391 | 0.7347 | 0.6444 | 0.7141 |
| | | | | | | Fused | 0.8162 | 0.8180 | 0.8381 | 0.7073 | 0.6810 | 0.7226 | 0.7119 | 0.6667 | 0.7279 |
| B | ✓ | ✗ | ✗ | ✗ | ✗ | Visual | 0.9071 | 0.9024 | 0.9248 | 0.8000 | 0.8020 | 0.8255 | 0.7838 | 0.7898 | 0.8333 |
| | | | | | | Audio | 0.8064 | 0.7630 | 0.8078 | 0.6964 | 0.6491 | 0.6562 | 0.7714 | 0.7333 | 0.7664 |
| | | | | | | Fused | 0.8462 | 0.8438 | 0.8628 | 0.7692 | 0.7708 | 0.7743 | 0.7647 | 0.7600 | 0.7886 |
| C | ✓ | ✓ | ✗ | ✗ | ✗ | Visual | 0.9375 | 0.9375 | 0.9635 | 0.9057 | 0.9123 | 0.9295 | 0.8816 | 0.8636 | 0.9289 |
| | | | | | | Audio | 0.8333 | 0.8160 | 0.8420 | 0.7292 | 0.6964 | 0.7236 | 0.8710 | 0.8750 | 0.8677 |
| | | | | | | Fused | 0.9167 | 0.9200 | 0.9143 | 0.8525 | 0.8772 | 0.8524 | 0.8421 | 0.8222 | 0.8494 |
| D | ✓ | ✓ | ✓ | ✗ | ✗ | Visual | 0.9600 | 0.9600 | 0.9948 | 0.9643 | 0.9649 | 0.9692 | 0.9333 | 0.9111 | 0.9551 |
| | | | | | | Audio | 0.8571 | 0.8333 | 0.8715 | 0.7463 | 0.7143 | 0.7659 | 0.9310 | 0.9111 | 0.9131 |
| | | | | | | Fused | 0.9375 | 0.9312 | 0.9248 | 0.9091 | 0.8958 | 0.9183 | 0.8621 | 0.8444 | 0.8919 |
| E | ✓ | ✓ | ✓ | ✓ | ✗ | Fused | 0.9479 | 0.9477 | 0.9419 | 0.9231 | 0.9167 | 0.9401 | 0.8909 | 0.8710 | 0.9037 |
| F | ✓ | ✓ | ✓ | ✗ | ✓ | Fused | 0.9408 | 0.9383 | 0.9455 | 0.9286 | 0.9200 | 0.9312 | 0.8814 | 0.8667 | 0.9146 |

## 6.6 CASE STUDY

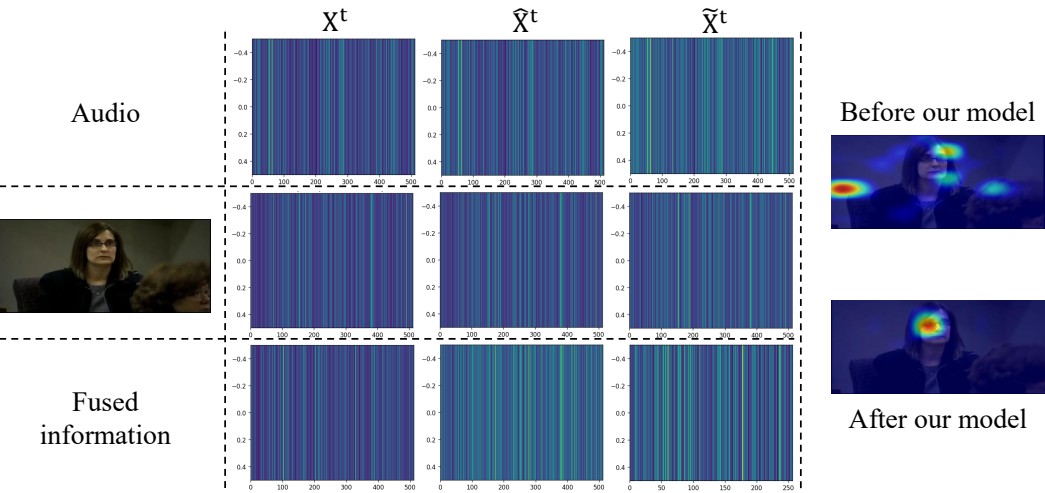

Figure 3: Interpretable Case Visualization.

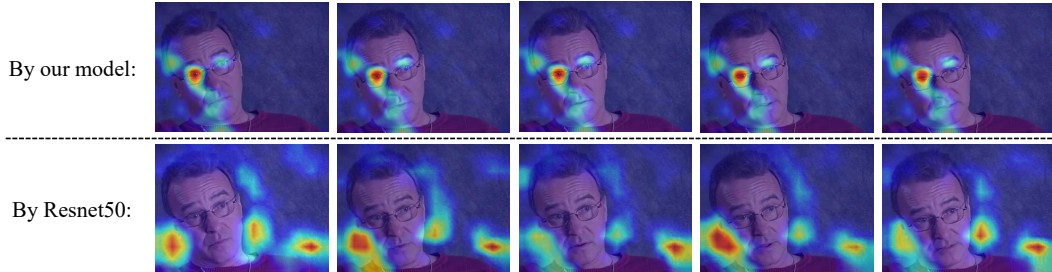

Figure 4: Visualization Examples of the Model's Attentional Regions.

To show how the model learns spectral representations and fusion, Fig. 3 visualizes features from the three modalities at three stages: the original state, after LCSF, and after OTCF. As training proceeds,

the feature distributions become more structured and support stronger discrimination. Early epochs show weakly differentiated spectra with near-uniform coloration; later epochs exhibit clearer band separation and greater variation in spectral energy.

A comparison of multimodal features before and after training is also provided. In Fig. 3, the untrained model attends to irrelevant regions, whereas the trained model focuses on facial areas, especially ocular motion, which is widely reported as a cue in deception detection.

To interpret model behavior, attention maps are visualized in Fig. 4. The model frequently attends to the pupils, a cue associated with deception through involuntary dilation, which aligns with domain reports. This contrasts with the irrelevant regions highlighted by a pre-trained ResNet50 and helps explain why ResNet-based baselines underperform on this task. Additional examples appear in Appendix Section H.

## 7 CONCLUSION

SPOT-JS is presented as a frequency-domain method for multimodal deception detection under domain shift. The system couples unified preprocessing and audiovisual synchronization (TDAM), a power-spectrumbased Learnable Chebyshev Spectrum Filter (LCSF), bidirectional fusion via entropy-regularized optimal transport (OTCF), and JS-Align for JensenShannonguided posterior matching. This design reduces reliance on invasive signals and handcrafted features, improves unimodal encodings, and provides principled multimodal alignment and fusion. A Chebyshev Spectrum-guided Knowledge Transfer (CSKT) module further transfers facial knowledge. Experiments show reduced dependence on traditional physiological cues and competitive accuracy across datasets.

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

## A    THE USE OF LARGE LANGUAGE MODELS (LLMS)

We employed a large language model solely for the purpose of polishing the written paragraphs to enhance their fluency and readability. Beyond this textual refinement, no other aspects of the work utilized large language models.

## B    ETHICS STATEMENT

As developers create deception detection systems, it must be a priority to uphold privacy, minimize psychological harm, and prevent discrimination. Potential misuse of such technology could impose significant negative impacts on society. If deployed without consent, these systems risk infringing on personal privacy by collecting and analyzing sensitive individual data-such as speech patterns, facial expressions, and body language-without adequate safeguards. Researchers must adhere to applicable regulations throughout the development and deployment of deception detection systems.

## C    CHEBYSHEV SPECTRUM-GUIDED KNOWLEDGE TRANSFER (CSKT) MODULE

Having obtained the fused multimodal representation $\mathbf{f}$, we provide further details of the Hierarchical Optimal Transport Knowledge Transfer (H-OTKT) componentJi et al. (2025), which serves as the foundation of our proposed CSKT module. H-OTKT transfers affective prior knowledge $\mathbf{X}^s$ from large-scale Visual Facial Expression Recognition (VFER) datasets (e.g., DFEW Jiang et al. (2020)) to enhance the discriminative power for deception detection.

Before performing knowledge transfer, we first employ Eq. (6) to transform the knowledge $\mathbf{X}^s$ to be transferred into the frequency domain. We then apply the LCSF module for filtering, thereby enabling more effective and informative feature transmission:

$$\mathbf{X}^{so} = \mathbf{LCSF}(\mathbf{X^s}) \tag{22}$$

where $\mathbf{LCSF}(\cdot)$ denotes the Learnable Chebyshev Spectrum Filter (LCSF) Module in our framework.

$\mathbf{f}$ is firstly mapped into $\mathbf{f}' = \mathcal{F}_1(\mathbf{f}) \in \mathbb{R}^{n \times d}$ by an MLP $\mathcal{F}_1$, such that the feature spaces between source and target domain could be the same, where $n$ is the batch size. Let $\mathcal{Q} = \sum_{k=1}^{L^s} \frac{1}{L^s} \delta_{\mathcal{Q}^k}$ as the discrete uniform distribution over $L^s$ classes of VFER dataset, $\mathcal{Q}^k$ is the representation vector of $k$-th class. And $\mathcal{P} = \sum_{i=1}^{n} \frac{1}{n} \delta_{\mathbf{f}_i'}$ is the discrete uniform distribution over $n$ target deception samples. Then, according to Eq. (10), the entropic regularized OT between $\mathcal{P}$ and $\mathcal{Q}$ is:

$$\mathrm{OT}_{high}(\mathcal{P}, \mathcal{Q}) = \min_{\mathbf{T} \in \Pi(\mathcal{P}, \mathcal{Q})} < \mathbf{T}, \mathbf{M} >_{\mathrm{F}} -\epsilon \mathcal{H}(\mathbf{T}) \tag{23}$$

where $\mathbf{T} \in \mathbb{R}^{n \times L^s}$ and $\mathbf{M} \in \mathbb{R}^{n \times L^s}$ are the transport plan and the cost matrix between facial expression classes and target deception samples. Each element $\mathbf{T}_{i,k}$ indicates the importance of the $k$-th class in VFER dataset for the $i$-th sample in deception mini-batch, determining which class and how much of knowledge should be transferred. Besides, $\mathbf{T}$ should satisfy the following constraint:

$$\Pi(\mathcal{P}, \mathcal{Q}) := \left\{ \sum_{i=1}^{n} \mathbf{T}_{i,k} = \frac{1}{L^s}, \ \sum_{k=1}^{L^s} \mathbf{T}_{i,k} = \frac{1}{n} \right\} \tag{24}$$

It is evident that the solution $\mathbf{T}$ relies on the cost matrix $\mathbf{M}$. Simply applying cosine similarity between the features of samples from a deception mini-batch and the mean features of each class in the VFER dataset may lead to a sub-optimal solution. Moreover, the contribution of different samples in each class may vary. Therefore, they adopt another optimal transport formulation to obtain the optimal $\mathbf{M}$. According to Guo et al. (2022), the empirical distribution of the $k$-th class is expressed as $\mathcal{Q}^k = \sum_{j=1}^{J_k} p_j^k \delta_{\mathbf{X}_j^{s,k}}$, where the importance $p_j^k$ of the $j$-th sample in the $k$-th source class is obtained by the logistic regression score. Based on this formulation, a low-level entropic regularized OT is further defined as follows:

$$\mathrm{OT}_{low}(\mathcal{P}, \mathcal{Q}^k) = \min_{\mathbf{T}^{low,k} \in \Pi(\mathcal{P}, \mathcal{Q}^k)} < \mathbf{T}^{low,k}, \mathbf{M}^{low,k} >_{\mathrm{F}} -\epsilon \mathcal{H}(\mathbf{T}^{low,k}) \tag{25}$$

$\Pi(\mathcal{P}, \mathcal{Q}^k) := \left\{ \sum_j^{J_k} \mathbf{T}_{i,j}^{low,k} p_j^k = \frac{1}{n}, \sum_i^n \mathbf{T}_{i,j}^{low,k} \frac{1}{n} = p_j^k \right\}$ is the constrain, and $\mathbf{T}^{low,k}$ is the transport plan between each sample in mini-batch and samples in the $k$-th source domain class. $\mathbf{M}^{low,k} \in \mathbb{R}^{n \times J_k}$ is determined by cosine similarity, $i.e.$, $\mathbf{M}_{i,j}^{low,k} = 1 - \cos(\mathbf{f}_i', \hat{\mathbf{X}}_j^s)$. The cost matrix $\mathbf{M}$ in high-level OT of Eq. (23) will be replaced by the total OT distance between each target deception sample and all sample in each class of VFER dataset, $i.e.$, $\mathbf{M}_{:,k} = <\mathbf{T}^{low,k}, \mathbf{M}^{low,k}>_{\text{F}}$.

For the optimization, both Eq. (23) and Eq. (25) are solved by Sinkhorn algorithm Cuturi (2013) hierarchically. Using the OT distance calculated from low-level OT as the cost $\mathbf{M}$ of high-level OT adaptively, CSKT is able to obtain the transport weight $\mathbf{T}$ between deception samples and facial expression classes, which is the potential correlation mapping of facial expression classes for target samples.

Once we obtained correlation mapping $\mathbf{T}$ by solving Eq. (23), knowledge transformation can be performed. For each sample in deception domain, more knowledge from highly associated classes should be transferred, while knowledge from uncorrelated classes should not be transferred. To realize it, the transferred knowledge $\mathbf{X}^{trans} \in \mathbb{R}^{n \times d}$ is represented as follows:

$$\mathbf{X}_i^{trans} = \mathcal{F}_2 \left( n \cdot \sum_{k=1}^{L^s} \mathbf{T}_{i,k} \left[ \frac{1}{J_k} \sum_{j=1}^{J_k} \hat{\mathbf{X}}_j^s \right] \right), \quad i = 1, \cdots, n \tag{26}$$

where $\frac{1}{J_k} \sum_{j=1}^{J_k} \hat{\mathbf{X}}_j^s$ denotes the average feature of samples belonging to the $k$-th class in source domain; $\mathbf{T}_{i,k}$ quantifies the correlation weight between the $k$-th source class and $i$-th deception sample; $n$ is used for scaling due to the constraint in high-level OT. And $\mathcal{F}_2$ is an MLP.

After obtaining $\mathbf{X}^{trans}$, we transform it back into the spatial domain using Eq. (14) for subsequent processing:

$$\mathbf{X}^{trans} \leftarrow \mathcal{F}_{seq}^{-1}(\mathbf{X}^{trans}) \tag{27}$$

In order to integrate the transferred knowledge $\mathbf{X}^{trans}$ with features $\mathbf{f}'$ extracted from target samples, the fused representation of deception detection samples are calculated as:

$$\mathbf{X}^{fused} = \xi' \mathbf{X}^{trans} + (1 - \xi') \mathbf{f}' \tag{28}$$

where $\xi'$ is the weight of transferred feature $\mathbf{X}^{trans}$. Since it's hard to learn excellent $\mathbf{f}'$ at the beginning of the training phase, a curriculum learning strategy (Kumar et al., 2010; Wang et al., 2021) is adopted as $\xi' = \frac{\xi}{2} \times \left( 1 - \cos \left( \frac{e-1}{N_e} \times \pi \right) \right)$, where $e$ is the current training epoch number and $N_e$ is the total training epoch number. As $\xi'$ is gradually increased, a better $\mathbf{f}'$ is gained for H-OTKT.

### C.1 Sample-specific Re-weighting Knowledge Bank (SRKB) Module

Since we do not make any modifications to the Sample-specific Re-weighting Knowledge Bank (SRKB) Module, we refer the reader to Ji et al. (2025) for further details, and do not elaborate on it here.

## D Detailed Description about Comparison Methods

In this section, we will give more detailed descriptions about the comparison methods.

### D.1 Traditional Machine Learning based Deception Detection Methods

Firstly, we would like to introduce the statistical features that are used in our experiments.

- Visual: OpenFace (Baltrusaitis et al., 2018) is an open source tool for extracting facial statistical features, such as landmarks, action units. Some of the action units also show high association with deception (Şen et al., 2020). Following the previous researches (Mathur & Matarić, 2020; Krishnamurthy et al., 2018; Avola et al., 2019; Yang et al., 2021a), we also employ OpenFace as our visual feature extractor to obtain visual statistical features.

- Audio: Mel-scale Frequency Cepstral Coefficients (MFCC) (Abdul & Al-Talabani, 2022) and OpenSMILE (Eyben et al., 2010) are two mostly used acoustic statistical features for detecting deception.

Not that the features extracted by OpenFace are frame-wise, since different video clips may contains different number of frames, we normalize the dimension of one video clip by OpenMM (Morales et al., 2017), which calculates the 11 statistical functionals for each feature at view label (Rill-García et al., 2019).

For classification, we employ SVM, Decision Tree Random Forest, and AdaBoost as our classifier. The statistical features are fed to each classifier to perform classification.

## D.2 Deep Learning based Deception Detection Methods

In this paper, we make comparisons with several deep learning methods. These methods can be separated by their backbone structure: Long Short-Term Memory (LSTM) (Graves & Graves, 2012) based, ResNet (He et al., 2016) based, and Transformer (Vaswani et al., 2017) based.

- LSTM based: LSTM is known as the sequence encoder, which is able to capture the contextual information of a given sequence. In this case, LSTM is employed to handle the contextual information aggregation at temporal dimension. Several researches adopt LSTM as the temporal encoder to obtain the temporal information (Mathur & Matarić, 2020; Krishnamurthy et al., 2018; Guo et al., 2023).

- ResNet based: ResNet is a common image encoder, which is built upon convolutional neural networks. In these works (Karnati et al., 2022; Ding et al., 2019a; Krishnamurthy et al., 2018; Guo et al., 2023), they use ResNet to automatically extract the visual features instead of using OpenFace or other manual approaches. In our experiments, ResNet with 18 layers (ResNet18) is employed to extract the visual features of each video frames.

- Transformer based: With the great success of Transformer (Vaswani et al., 2017), encoders with more parameters based on Transformer architecture have been proposed to encode video clips or audio sequences automatically with rich semantic information. W2V2 (Baevski et al., 2020) is typical audio encoder based on Transformer architecture, and VideoMAE (Tong et al., 2022) is able to directly encode the given video clip to a fixed length vector.

In our experiment, we make comparisons with the following researches.

- ResNet18 + LSTM (Karnati et al., 2022; Ding et al., 2019a; Krishnamurthy et al., 2018; Guo et al., 2023): In these methods, ResNet18 was adopted to extract video frame features of a video. Then sequential information of all frame features was formulated by an LSTM. Then an MLP performed classification using the last output feature of the sequence.

- W2V2 + MLP (Guo et al., 2023; Karnati et al., 2022; Krishnamurthy et al., 2018): In these methods, W2V2 model was used to extract audio features. Then an MLP was used to make classification.

- ResNet18 ⊕ OpenSMILE (Gogate et al., 2017; Krishnamurthy et al., 2018; Guo et al., 2023): These methods took both visual and audio modalities into account, and performed late fusion from each single modality branch.

- Face frames+FFCSN (Ding et al., 2019b): These methods address the challenge of deception detection in unconstrained videos. To facilitate joint deep feature learning from facial expressions and body movements, a Facial-Focused Cross-Stream Network (FFCSN) was proposed to handle the temporal misalignment between these cues. Additionally, meta-learning and adversarial learning were incorporated into the model training framework.

## D.3 Transfer Learning based Methods

There have been a small number of researches that tried to transfer knowledge from other related dataset to enhance the detection performance with deep learning based methods. Therefore, we adapt several common kinds of transfer learning strategy to the deception detection task.

- Optimal Transport based: Free Lunch (Yang et al., 2021b) achieved knowledge transfer by estimating the weight of each base class and perform distribution calibration with the statistics of base classes, which directly used the distance of class average feature and support feature as the measurement. Similar to FreeLunch, ADC (Guo et al., 2022) also aimed to transfer knowledge via quantifying the weight of each source class and target sample and perform distribution calibration. The optimal transport plan represents the importance (or correlation) between the base classes and the novel samples. In AFFAKT (Ji et al., 2025), the relation between deception sample and each facial expression class is estimated by these two methods in the forward process, which play roles with the H-OTKT and SRKB modules.

- Pre-train & Fine-tune based: The transfer learning methods of this kind are more likely to be adopted in large models, such as PECL (Guo et al., 2023). It tried to transfer knowledge from the pre-trained dataset and checkpoint to the target dataset.

- Knowledge Distillation based: Knowledge distillation is also known as a typical transfer learning method. In (Gu et al., 2023), knowledge distillation was used for discover novel class samples given a model pre-trained on a source dataset. The key idea of (Gu et al., 2023) is to distill knowledge according to the class realtion.

## D.4 Fusion Methods

Fusion methods employ various approaches for multimodal integration. Channel-wise feature concatenation (Concat) provides lightweight fusion, while SE-Concat enhances this process through squeeze-and-excitation modules (Hu et al., 2018) for modality-specific refinement. CMFL (George & Marcel, 2021) introduces adaptive channel weighting via cross-modal focal loss. The Prompt method (Jia et al., 2022) learns task-specific visual tokens while keeping transformer blocks frozen. AVA (Li et al., 2024) achieves synchronized feature integration by aligning temporal embeddings across visual and auditory modalities.

- SE-Concat: Squeeze-and-excitation(SE) module is utilized in each independent modality branch first. With the channel-wise self-calibration via the SE module, the refined features are then concatenated.

- CMFL: Cross-modal focal loss is used to modulate the loss contribution of each channel as a function of the confidence of individual channels.

- Prompt: Visual Prompt Tuning method introduces a small amount of task-specific learnable tokens while freezing the entire pretrained transformer blocks during deception detection training.

- AVA: This work introduces a novel Transformer-based framework incorporating Audio-Visual Adapter modules and Cross Uni- and Fused Modal Contrastive Loss (CUFMCL) for multi-modal deception detection which achieves superior performance under flexible-modal scenarios.

## E  More experiments and The Standard Deviation Report Between Folds

As discussed in the main text, we report here all cross-domain testing results along with more detailed experimental findings. As shown in Tables 7 and 8, we additionally present the remaining results for F1 and AUC. The average value and the standard deviation between different folds are shown in Tables 5 and 6. Beside the analysis in the main text, the results in Tables 5 and 6 show that SPOT-JS is more robust, since the evaluate metric between different folds have smaller standard deviation value.

## F  More Ablation Studies

In this section, we systematically conduct ablation studies on all the proposed modules to thoroughly demonstrate the effectiveness of each component. For clarity, when the CSKT module is excluded, the original H-OTKT module is used instead. Likewise, when the OTCF or JS-Align modules are

Table 5: Comparison results on RLT dataset and DOLOS dataset with F1, ACC and AUC metrics. Both mean and standard deviation are reported (mean±std).

(a) Results with visual modality.

| Target | RLT | | | DOLOS | | |
|---|---|---|---|---|---|---|
| Method | F1 | ACC | AUC | F1 | ACC | AUC |
| OpenFace + SVM | 0.2253±0.2605 | 0.5293±0.0361 | 0.5571±0.0470 | 0.6975±0.0010 | 0.5355±0.0012 | 0.5430±0.0160 |
| OpenFace + Decision Tree | 0.5553±0.1157 | 0.5303±0.1048 | 0.5303±0.1048 | 0.5358±0.0303 | 0.5058±0.0262 | 0.5058±0.0262 |
| OpenFace + Random Forest | 0.6033±0.0867 | 0.6033±0.0559 | 0.5997±0.0574 | 0.6175±0.0193 | 0.5367±0.0227 | 0.5466±0.0272 |
| OpenFace + AdaBoost | 0.5199±0.1523 | 0.5303±0.0980 | 0.5766±0.1070 | 0.5536±0.0251 | 0.5057±0.0329 | 0.5035±0.0357 |
| AU + SVM | 0.4562±0.0723 | 0.5043±0.0726 | 0.4670±0.0970 | 0.6813±0.0194 | 0.5276±0.0126 | 0.5242±0.0089 |
| AU + Decision Tree | 0.4466±0.1577 | 0.4643±0.1167 | 0.4643±0.1167 | 0.5453±0.0172 | 0.5173±0.0137 | 0.5173±0.0137 |
| AU + Random Forest | 0.5534±0.0792 | 0.5463±0.0810 | 0.5330±0.0766 | 0.5808±0.0183 | 0.5045±0.0256 | 0.5157±0.0230 |
| AU + AdaBoost | 0.5130±0.0530 | 0.4877±0.0612 | 0.4835±0.0833 | 0.5295±0.0302 | 0.4876±0.0185 | 0.4735±0.0264 |
| OpenFace + LSTM | 0.5241±0.0995 | 0.5623±0.0834 | 0.5952±0.1164 | 0.5928±0.0342 | 0.5628±0.0164 | 0.5854±0.0152 |
| AU + LSTM | 0.4888±0.0472 | 0.6197±0.0419 | 0.6760±0.0442 | 0.6343±0.0084 | 0.5646±0.0137 | 0.5868±0.0098 |
| ResNet18 + LSTM | 0.4996±0.1391 | 0.6117±0.0718 | 0.6387±0.0928 | 0.6415±0.0124 | 0.5972±0.0087 | 0.5668±0.0136 |
| PECL(only visual) | 0.5880±0.1018 | 0.6528±0.0040 | 0.6734±0.0508 | 0.7010±0.0213 | 0.6387±0.0139 | 0.6770±0.0099 |
| FreeLunch | 0.7612±0.1207 | 0.8090±0.0782 | 0.8712±0.0782 | 0.6961±0.0147 | 0.6222±0.0221 | 0.6444±0.0221 |
| ADC | 0.7793±0.1218 | 0.8173±0.0942 | 0.8674±0.0943 | 0.6880±0.0163 | 0.6716±0.0157 | 0.7206±0.0157 |
| Cr-KD-NCD | 0.6957±0.1342 | 0.7200±0.0869 | 0.6928±0.0865 | 0.5850±0.0175 | 0.6091±0.0186 | 0.6013±0.0202 |
| AFFAKT | 0.8760±0.0516 | 0.8670±0.0558 | 0.8789±0.0516 | 0.7102±0.0233 | 0.6764±0.0199 | 0.7212±0.0292 |
| SPOT-JS(Ours) | **0.9600±0.0534** | **0.9600±0.0406** | **0.9948±0.0406** | **0.9643±0.0348** | **0.9649±0.0299** | **0.9692±0.0299** |
| | +8.40% | +9.30% | +11.59% | +25.41% | +28.85% | +24.80% |

(b) Results with audio modality.

| Target | RLT | | | DOLOS | | |
|---|---|---|---|---|---|---|
| Method | F1 | ACC | AUC | F1 | ACC | AUC |
| MFCC + MLP | 0.5226±0.2911 | 0.6367±0.1263 | 0.7030±0.0502 | 0.5963±0.0757 | 0.5810±0.0232 | 0.6134±0.0279 |
| OpenSMILE + MLP | 0.6885±0.1275 | 0.6597±0.1121 | 0.5926±0.0916 | 0.6867±0.0128 | 0.5537±0.0095 | 0.5325±0.0091 |
| W2V2 + MLP | 0.6117±0.0810 | 0.6780±0.0266 | 0.6106±0.0631 | 0.4383±0.0333 | 0.5421±0.0115 | 0.5369±0.0120 |
| PECL(only audio) | 0.7121±0.0748 | 0.7100±0.0718 | 0.6962±0.0796 | 0.6777±0.0364 | 0.6119±0.0200 | 0.6281±0.0155 |
| FreeLunch | 0.6432±0.0989 | 0.6850±0.0704 | 0.6944±0.0704 | 0.6589±0.0334 | 0.5979±0.0203 | 0.6196±0.0188 |
| ADC | 0.6402±0.0922 | 0.6767±0.0744 | 0.6858±0.0744 | 0.6196±0.0814 | 0.6058±0.0135 | 0.6040±0.0135 |
| AFFAKT | 0.7316±0.0493 | 0.7440±0.0707 | 0.7396±0.0768 | 0.6982±0.0210 | 0.6198±0.0076 | 0.6391±0.0184 |
| SPOT-JS(Ours) | **0.8571±0.0222** | **0.8333±0.0161** | **0.8438±0.0161** | **0.7463±0.0214** | **0.7143±0.0094** | **0.7659±0.0094** |
| | +12.55% | +8.93% | +13.19% | +4.81% | +9.45% | +12.68% |

(c) Results with fused modalities.

| Target | RLT | | | DOLOS | | |
|---|---|---|---|---|---|---|
| Method | F1 | ACC | AUC | F1 | ACC | AUC |
| OpenFace ⊕ OpenSMILE | 0.6895±0.0463 | 0.6781±0.0752 | 0.6212±0.0671 | 0.6124±0.0354 | 0.5986±0.0153 | 0.5863±0.0136 |
| ResNet18 ⊕ OpenSMILE | 0.6283±0.0498 | 0.6853±0.0627 | 0.6598±0.0763 | 0.5863±0.0263 | 0.6152±0.0175 | 0.6485±0.0121 |
| PECL | 0.7102±0.0215 | 0.6939±0.0488 | 0.7424±0.0569 | 0.7084±0.0142 | 0.6597±0.0114 | 0.6353±0.0108 |
| FreeLunch | 0.7695±0.0799 | 0.8093±0.0782 | 0.8547±0.0781 | 0.6807±0.0251 | 0.6289±0.0060 | 0.6669±0.0060 |
| ADC | 0.7493±0.0703 | 0.8093±0.0782 | 0.8446±0.0782 | 0.6997±0.0010 | 0.6746±0.0126 | 0.7307±0.0115 |
| AFFAKT | 0.8412±0.0848 | 0.8427±0.0768 | 0.8563±0.0688 | 0.7149±0.0099 | 0.6810±0.0140 | 0.7289±0.0092 |
| SPOT-JS(Ours) | **0.9630±0.0473** | **0.9600±0.0172** | **0.9679±0.0172** | **0.9474±0.0264** | **0.9474±0.0233** | **0.9846±0.0233** |
| | +12.18% | +11.73% | +11.16% | +23.25% | +26.64% | +25.57% |

absent, we adopt a simple baseline strategy that sums half of the video features with half of the audio features. When TDAM is not used, we adopt the traditional method of encoding each image individually. The results are shown in Table 9.

# G  DATASETS AND EXPERIMENTAL SETTINGS

## G.1  DATASETS

**Deception Detection Datasets.** We conduct the experiments on three most widely used datasets in deception detection task, Real Life Trial (RLT) dataset, DOLOS dataset and Box of Lies (BOL) dataset:

- **Real Life Trial (RLT)** dataset is a popular real-world dataset collected from public court trials, which consists of 121 videos including 61 deceptive and 60 truthful video clips. As it is a real-world dataset, the Real Life Trial dataset has more noise on both the video and audio.

Table 6: Comparison results on BOL dataset with F1, ACC and AUC metrics. Both mean and standard deviation are reported (mean±std).

(a) Results with visual modalities.

| Target | Box of lies | | |
|---|---|---|---|
| Method | F1 | ACC | AUC |
| CMFL | 0.5584±0.0508 | 0.4403±0.0607 | 0.4907±0.0512 |
| SE-Concat | 0.5606±0.0613 | 0.5678±0.0577 | 0.5657±0.0546 |
| Prompt | 0.6785±0.0512 | 0.5451±0.0566 | 0.6143±0.0483 |
| PECL | 0.6832±0.0486 | 0.5705±0.0511 | 0.6476±0.0431 |
| AVA+CUFMCL | 0.6953±0.0712 | 0.5947±0.0733 | 0.6743±0.0789 |
| SPOT-JS(Ours) | **0.9333±0.0066** | **0.9111±0.0088** | **0.9551±0.0088** |
| | **+23.80%** | **+31.64%** | **+28.08%** |

(b) Results with audio modalities.

| Target | Box of lies | | |
|---|---|---|---|
| Method | F1 | ACC | AUC |
| CMFL | 0.5812±0.0723 | 0.5308±0.0755 | 0.5406±0.0637 |
| SE-Concat | 0.5694±0.0688 | 0.5530±0.0639 | 0.5638±0.0628 |
| Prompt | 0.6684±0.0433 | 0.5673±0.0487 | 0.5799±0.0453 |
| PECL | 0.6726±0.0311 | 0.5828±0.0343 | 0.6122±0.0289 |
| AVA+CUFMCL | 0.6972±0.0233 | 0.5987±0.0289 | 0.6456±0.0241 |
| SPOT-JS(Ours) | **0.9310±0.0066** | **0.9111±0.0088** | **0.9131±0.0088** |
| | **+23.38%** | **+31.24%** | **+26.75%** |

(c) Results with fused modalities.

| Target | Box of lies | | |
|---|---|---|---|
| Method | F1 | ACC | AUC |
| CMFL | 0.6568±0.0763 | 0.5350±0.0688 | 0.5635±0.0873 |
| SE-Concat | 0.6721±0.0733 | 0.5919±0.0782 | 0.6109±0.0725 |
| Prompt | 0.6891±0.0655 | 0.5954±0.0635 | 0.6256±0.0689 |
| PECL | 0.6723±0.0543 | 0.6078±0.0509 | 0.6433±0.0578 |
| AVA+CUFMCL | 0.6920±0.0482 | 0.6256±0.0473 | 0.6667±0.0479 |
| SPOT-JS(Ours) | **0.9153±0.0163** | **0.8889±0.0243** | **0.9393±0.0243** |
| | **+22.33%** | **+26.33%** | **+27.26%** |

- **Box of Lies (BOL)** is a deception dataset collected from an online gameshow, which consists of 225 videos including 144 deceptive and 81 truthful video clips. (6 male and 20 female). The full video set contains 29 truthful and 36 deceptive rounds of games.

- **DOLOS** is the largest game-show deception detection dataset recently proposed in the field, containing rich deceptive dialogues. The dataset consists of 1,675 video clips featuring 213 subjects (141 male and 72 female participants), with each clip lasting 2-19 seconds.

Please note that the DOLOS dataset is not a publicly available dataset. If you wish to use this dataset, you need to submit a relevant application.

**Facial Expression Recognition Datasets.** One *in-the-wild* VFER datasets (DFEW is employed in our experiments. It contains 16372 samples with 7 expression categories. For DFEW, we only use 11697 single-labeled clips:

- **DFEW** is a large-scale real-world dataset collected from over 1,500 movies, consisting of 16,372 video clips annotated with seven basic emotions (anger, disgust, fear, happy, sad, surprise, neutral). As a real-world dataset, DFEW contains significant variations in illumination, pose, and occlusion, making it highly challenging for facial expression recognition tasks.

If you wish to use this dataset, you need to submit a relevant application.

Table 7: We report cross-domain experimental results on the RLT(R), BOL(B), and DOLOS(D) datasets using the **F1** metric. Here, Ff refers to Face frames, while Mel refers to Mel spectrograms.

(a) Results with visual modality.

| Method | R to D | D to B | R to B | D to R | B to R | B to D |
|---|---|---|---|---|---|---|
| AU+LSTM | 0.4983 | 0.4979 | 0.5876 | 0.5322 | 0.5456 | 0.5124 |
| Gaze+MLP | 0.4869 | 0.5111 | 0.5948 | 0.5376 | 0.5569 | 0.5103 |
| AU+Gaze+MLP | 0.5089 | 0.5107 | 0.6569 | 0.5488 | 0.5516 | 0.5123 |
| Affect+MLP | 0.5186 | 0.5132 | 0.5869 | 0.5427 | 0.5539 | 0.5213 |
| AU+Gaze+Affect+MLP | 0.5257 | 0.5122 | 0.5947 | 0.5529 | 0.5563 | 0.5273 |
| Ff+ResNet18 | 0.5283 | 0.5107 | 0.6109 | 0.5423 | 0.5674 | 0.5239 |
| Ff+ResNet18+GRU | 0.5238 | 0.5228 | 0.6329 | 0.5483 | 0.5613 | 0.5284 |
| Ff+ResNet18+KNN | 0.5329 | 0.5137 | 0.6307 | 0.5544 | 0.5689 | 0.5328 |
| Ff+ResNet18+SVM | 0.5428 | 0.5102 | 0.6218 | 0.5628 | 0.5719 | 0.5374 |
| Ff+FFCSN | 0.5328 | 0.5237 | 0.6279 | 0.5695 | 0.5673 | 0.5478 |
| CLBF | 0.5334 | 0.5134 | 0.4508 | 0.5434 | 0.5739 | 0.5428 |
| PECL | 0.5616 | 0.5337 | 0.6329 | 0.5528 | 0.5289 | 0.5548 |
| SPOT-JS(Ours) | **0.7188** | **0.8438** | **0.9333** | **0.7500** | **0.7059** | **0.6588** |
| | **+15.72%** | **+31.01%** | **+27.64%** | **+18.05%** | **+13.20%** | **+10.40%** |

(b) Results with audio modality.

| Method | R to D | D to B | R to B | D to R | B to R | B to D |
|---|---|---|---|---|---|---|
| Acoustic + Prosodic+MLP | 0.4537 | 0.5127 | 0.5319 | 0.5123 | 0.5226 | 0.4989 |
| Mel+ResNet18 | 0.4989 | 0.5329 | 0.5384 | 0.5408 | 0.5279 | 0.4997 |
| Mel+ResNet18+KNN | 0.4864 | 0.5278 | 0.5467 | 0.5396 | 0.5431 | 0.5024 |
| Mel+ResNet18+SVM | 0.4927 | 0.5428 | 0.5427 | 0.5489 | 0.5487 | 0.5126 |
| Waveform+Wave2Vec | 0.5123 | 0.5319 | 0.5029 | 0.5499 | 0.5389 | 0.5183 |
| PECL | 0.5233 | 0.5307 | 0.5489 | 0.6071 | 0.5667 | 0.5237 |
| SPOT-JS(Ours) | **0.6400** | **0.7541** | **0.8333** | **0.6667** | **0.7586** | **0.6588** |
| | **+11.67%** | **+21.13%** | **+28.44%** | **+5.96%** | **+19.19%** | **+13.51%** |

(c) Results with fused modality.

| Method | R to D | D to B | R to B | D to R | B to R | B to D |
|---|---|---|---|---|---|---|
| Average | 0.5394 | 0.5843 | 0.5894 | 0.5407 | 0.4926 | 0.5128 |
| Concat | 0.5488 | 0.5869 | 0.5913 | 0.5659 | 0.4937 | 0.5146 |
| SE-Concat | 0.5349 | 0.5987 | 0.6157 | 0.5683 | 0.5127 | 0.5183 |
| Cross-Atten | 0.5429 | 0.6017 | 0.6169 | 0.5716 | 0.5186 | 0.5247 |
| MLP-Mixer | 0.5517 | 0.6149 | 0.5789 | 0.5783 | 0.5273 | 0.5383 |
| PECL | 0.5636 | 0.6192 | 0.5983 | 0.5639 | 0.5473 | 0.5473 |
| Atten-Mixer | 0.5689 | 0.6497 | 0.6239 | 0.5938 | 0.5429 | 0.5409 |
| SPOT-JS(Ours) | **0.6910** | **0.8125** | **0.8358** | **0.7500** | **0.8889** | **0.6506** |
| | **+12.21%** | **+16.28%** | **+21.29%** | **+15.62%** | **+34.16%** | **+10.33%** |

## G.2 SPECIFIC EXPERIMENTAL DETAILS

Our experiments were conducted on a system running Ubuntu 22.04 and Python 3.9 with Torch 2.7.0, utilizing one RTX 4090 24GB GPU. All experiments share the same configuration: a threshold of $1 \times 10^{-5}$, a learning rate of $1 \times 10^{-5}$, $\alpha$ set to 0.95, $\delta$ set to 0.01, $\epsilon$ set to 0.01, and the AdamW optimizer with a weight decay of $1 \times 10^{-5}$. The batch size is set to 16 for video and fused modalities, and 64 for the audio modality during the training phase. For the testing phase, the batch size is 2 across all datasets and modalities, with 5-fold cross-validation. For a detailed explanation of the parameter symbols, please refer to Section G.3.

For the RLT dataset, the hyperparameters are configured as follows: For the video modality, we use $\xi$ set to 0.2 and $\nu$ set to 0.1, with training for 20 epochs. The audio modality employs $\xi$ set to 0.5 and

Table 8: We report cross-domain experimental results on the RLT(R), BOL(B), and DOLOS(D) datasets using the **AUC** metric. Here, Ff refers to Face frames, while Mel refers to Mel spectrograms.

(a) Results with visual modality.

| Method | R to D | D to B | R to B | D to R | B to R | B to D |
|---|---|---|---|---|---|---|
| AU+LSTM | 0.5017 | 0.5042 | 0.5776 | 0.5327 | 0.5437 | 0.5027 |
| Gaze+MLP | 0.5063 | 0.5093 | 0.5889 | 0.5317 | 0.5536 | 0.5113 |
| AU+Gaze+MLP | 0.5173 | 0.5147 | 0.6413 | 0.5376 | 0.5482 | 0.5174 |
| Affect+MLP | 0.5143 | 0.5129 | 0.5836 | 0.5568 | 0.5513 | 0.5238 |
| AU+Gaze+Affect+MLP | 0.5243 | 0.5078 | 0.5917 | 0.5463 | 0.5523 | 0.5217 |
| Ff+ResNet18 | 0.5216 | 0.5144 | 0.6167 | 0.5489 | 0.5673 | 0.5289 |
| Ff+ResNet18+GRU | 0.5233 | 0.5273 | 0.6326 | 0.5583 | 0.5603 | 0.5311 |
| Ff+ResNet18+KNN | 0.5319 | 0.5198 | 0.6337 | 0.5519 | 0.5617 | 0.5344 |
| Ff+ResNet18+SVM | 0.5397 | 0.5089 | 0.6343 | 0.5609 | 0.5779 | 0.5333 |
| Ff+FFCSN | 0.5361 | 0.5219 | 0.6128 | 0.5697 | 0.5571 | 0.5421 |
| CLBF | 0.5217 | 0.5136 | 0.4409 | 0.5476 | 0.5923 | 0.5567 |
| PECL | 0.5416 | 0.5289 | 0.6517 | 0.5573 | 0.5783 | 0.5538 |
| SPOT-JS(Ours) | **0.7063** | **0.8410** | **0.9437** | **0.7726** | **0.7344** | **0.6799** |
| | **+16.47%** | **+31.21%** | **+29.20%** | **+20.29%** | **+14.21%** | **+12.32%** |

(b) Results with audio modality.

| Method | R to D | D to B | R to B | D to R | B to R | B to D |
|---|---|---|---|---|---|---|
| Acoustic + Prosodic+MLP | 0.4543 | 0.5139 | 0.5228 | 0.5117 | 0.5179 | 0.4928 |
| Mel+ResNet18 | 0.5112 | 0.5216 | 0.5333 | 0.5233 | 0.5233 | 0.4969 |
| Mel+ResNet18+KNN | 0.4839 | 0.5276 | 0.5444 | 0.5322 | 0.5356 | 0.4822 |
| Mel+ResNet18+SVM | 0.4924 | 0.5389 | 0.5416 | 0.5446 | 0.5369 | 0.5111 |
| Waveform+Wave2Vec | 0.5073 | 0.5372 | 0.4907 | 0.5333 | 0.5377 | 0.5123 |
| PECL | 0.5236 | 0.5366 | 0.5413 | 0.5726 | 0.5499 | 0.5169 |
| SPOT-JS(Ours) | **0.6257** | **0.7447** | **0.7946** | **0.6927** | **0.6719** | **0.6316** |
| | **+10.21%** | **+20.58%** | **+25.02%** | **+12.01%** | **+12.20%** | **+11.47%** |

(c) Results with fused modality.

| Method | R to D | D to B | R to B | D to R | B to R | B to D |
|---|---|---|---|---|---|---|
| Average | 0.5329 | 0.5843 | 0.5864 | 0.5343 | 0.4927 | 0.5128 |
| Concat | 0.5389 | 0.5867 | 0.5837 | 0.5639 | 0.4993 | 0.5143 |
| SE-Concat | 0.5403 | 0.6013 | 0.6133 | 0.5647 | 0.5103 | 0.5162 |
| Cross-Atten | 0.5489 | 0.5986 | 0.6176 | 0.5761 | 0.5123 | 0.5197 |
| MLP-Mixer | 0.5512 | 0.6007 | 0.5786 | 0.5729 | 0.5219 | 0.5286 |
| PECL | 0.5593 | 0.6129 | 0.5943 | 0.5623 | 0.5362 | 0.5389 |
| Atten-Mixer | 0.5673 | 0.6217 | 0.6113 | 0.5899 | 0.5346 | 0.5409 |
| SPOT-JS(Ours) | **0.7178** | **0.7580** | **0.7941** | **0.7326** | **0.8768** | **0.6000** |
| | **+15.05%** | **+13.63%** | **+17.65%** | **+14.27%** | **+34.06%** | **+5.91%** |

$\nu$ set to 0.05, also trained for 20 epochs. The fused modality adopts $\xi$ set to 0.2, $\nu$ set to 0.05, with training extended to 25 epochs.

For the DOLOS dataset: the video modality is configured with $\xi$ set to 0.2 and $\nu$ set to 0.1, trained for 25 epochs; the audio modality uses $\xi$ set to 0.25 and $\nu$ set to 0.1, trained for 25 epochs; and the fused modality employs $\xi$ set to 0.2, $\nu$ set to 0.1, trained for 25 epochs.

For the BOL dataset: the video modality parameters include $\xi$ set to 0.4 and $\nu$ set to 0.05, trained for 25 epochs; the audio modality uses $\xi$ set to 0.2 and $\nu$ set to 0.1, trained for 25 epochs; and the fused modality is configured with $\xi$ set to 0.2, $\nu$ set to 0.1, trained for 25 epochs.

Table 9: Ablation studies results. ① TDAM module, ② LCSF module, ③ CSKT module, ④ OTCF module, ⑤ JS-Align module.

| Case | ① | ② | ③ | ④ | ⑤ | modality | RLT F1 | RLT ACC | RLT AUC | DOLOS F1 | DOLOS ACC | DOLOS AUC | BOL F1 | BOL ACC | BOL AUC |
|---|---|---|---|---|---|---|---|---|---|---|---|---|---|---|---|
| A | ✗ | ✗ | ✗ | ✗ | ✗ | Visual | 0.8760 | 0.8670 | 0.8789 | 0.7054 | 0.6764 | 0.7212 | 0.7302 | 0.6889 | 0.7377 |
|  |  |  |  |  |  | Audio | 0.7267 | 0.7270 | 0.7218 | 0.6822 | 0.6198 | 0.6391 | 0.7347 | 0.6444 | 0.7141 |
|  |  |  |  |  |  | Fused | 0.8162 | 0.8180 | 0.8381 | 0.7073 | 0.6810 | 0.7226 | 0.7119 | 0.6667 | 0.7279 |
| A1 | ✗ | ✗ | ✗ | ✓ | ✗ | Fused | 0.8333 | 0.8352 | 0.8594 | 0.7200 | 0.6927 | 0.7536 | 0.7407 | 0.7083 | 0.7551 |
| A2 | ✗ | ✗ | ✗ | ✗ | ✓ | Fused | 0.8364 | 0.8393 | 0.8435 | 0.7213 | 0.6920 | 0.7552 | 0.7479 | 0.7045 | 0.7560 |
| A3 | ✗ | ✗ | ✗ | ✓ | ✓ | Fused | 0.8406 | 0.8444 | 0.8628 | 0.7576 | 0.7045 | 0.9312 | 0.7755 | 0.7216 | 0.7913 |
| B | ✓ | ✗ | ✗ | ✗ | ✗ | Visual | 0.9071 | 0.9024 | 0.9248 | 0.8000 | 0.8020 | 0.8255 | 0.7838 | 0.7898 | 0.8333 |
|  |  |  |  |  |  | Audio | 0.8064 | 0.7630 | 0.8078 | 0.6964 | 0.6491 | 0.6562 | 0.7714 | 0.7333 | 0.7664 |
|  |  |  |  |  |  | Fused | 0.8462 | 0.8438 | 0.8628 | 0.7692 | 0.7708 | 0.7743 | 0.7647 | 0.7600 | 0.7886 |
| B1 | ✓ | ✗ | ✗ | ✓ | ✗ | Fused | 0.8788 | 0.8750 | 0.8928 | 0.7937 | 0.7917 | 0.8158 | 0.7945 | 0.7935 | 0.8160 |
| B2 | ✓ | ✗ | ✗ | ✗ | ✓ | Fused | 0.8772 | 0.8800 | 0.8944 | 0.7925 | 0.7902 | 0.8198 | 0.8000 | 0.7975 | 0.8143 |
| B3 | ✓ | ✗ | ✗ | ✓ | ✓ | Fused | 0.8929 | 0.8977 | 0.9110 | 0.8182 | 0.8068 | 0.8499 | 0.8333 | 0.8240 | 0.8333 |
| C | ✗ | ✓ | ✗ | ✗ | ✗ | Visual | 0.8929 | 0.8854 | 0.8912 | 0.7600 | 0.7500 | 0.7777 | 0.7917 | 0.7586 | 0.8000 |
|  |  |  |  |  |  | Audio | 0.7797 | 0.7777 | 0.7761 | 0.6910 | 0.6364 | 0.6545 | 0.7500 | 0.7216 | 0.7760 |
|  |  |  |  |  |  | Fused | 0.8485 | 0.8519 | 0.8889 | 0.7407 | 0.7216 | 0.7630 | 0.7600 | 0.7102 | 0.7708 |
| C1 | ✗ | ✓ | ✗ | ✓ | ✗ | Fused | 0.8523 | 0.8693 | 0.8928 | 0.7502 | 0.7368 | 0.7743 | 0.7708 | 0.7333 | 0.7857 |
| C2 | ✗ | ✓ | ✗ | ✗ | ✓ | Fused | 0.8593 | 0.8600 | 0.8957 | 0.7536 | 0.7378 | 0.7785 | 0.7786 | 0.7395 | 0.7814 |
| C3 | ✗ | ✓ | ✗ | ✓ | ✓ | Fused | 0.8715 | 0.8767 | 0.9063 | 0.7605 | 0.7429 | 0.7857 | 0.7863 | 0.7501 | 0.7969 |
| D | ✗ | ✗ | ✓ | ✗ | ✗ | Visual | 0.9050 | 0.8977 | 0.8933 | 0.7760 | 0.7692 | 0.7976 | 0.7954 | 0.7981 | 0.8182 |
|  |  |  |  |  |  | Audio | 0.7692 | 0.7763 | 0.7857 | 0.6987 | 0.6506 | 0.6588 | 0.8523 | 0.8514 | 0.8660 |
|  |  |  |  |  |  | Fused | 0.8462 | 0.8438 | 0.8628 | 0.7692 | 0.7708 | 0.7743 | 0.7647 | 0.7600 | 0.7886 |
| D1 | ✗ | ✗ | ✓ | ✓ | ✗ | Fused | 0.8580 | 0.8571 | 0.8785 | 0.7763 | 0.7814 | 0.7891 | 0.7857 | 0.7812 | 0.7981 |
| D2 | ✗ | ✗ | ✓ | ✗ | ✓ | Fused | 0.8609 | 0.8593 | 0.8766 | 0.7786 | 0.7784 | 0.7857 | 0.7785 | 0.7796 | 0.7934 |
| D3 | ✗ | ✗ | ✓ | ✓ | ✓ | Fused | 0.8818 | 0.8800 | 0.8912 | 0.7898 | 0.7917 | 0.8021 | 0.7990 | 0.7921 | 0.8191 |
| E | ✓ | ✓ | ✗ | ✗ | ✗ | Visual | 0.9375 | 0.9375 | 0.9635 | 0.9057 | 0.9123 | 0.9295 | 0.8816 | 0.8636 | 0.9289 |
|  |  |  |  |  |  | Audio | 0.8333 | 0.8160 | 0.8420 | 0.7292 | 0.6964 | 0.7236 | 0.8710 | 0.8750 | 0.8677 |
|  |  |  |  |  |  | Fused | 0.9167 | 0.9200 | 0.9143 | 0.8525 | 0.8772 | 0.8524 | 0.8421 | 0.8222 | 0.8494 |
| E1 | ✓ | ✓ | ✗ | ✓ | ✗ | Fused | 0.9286 | 0.9261 | 0.9253 | 0.8848 | 0.8854 | 0.8848 | 0.8571 | 0.8370 | 0.8736 |
| E2 | ✓ | ✓ | ✗ | ✗ | ✓ | Fused | 0.9231 | 0.9271 | 0.9248 | 0.8864 | 0.8809 | 0.8928 | 0.8529 | 0.8321 | 0.8770 |
| E3 | ✓ | ✓ | ✗ | ✓ | ✓ | Fused | 0.9333 | 0.9323 | 0.9306 | 0.9024 | 0.8977 | 0.9120 | 0.8696 | 0.8524 | 0.8928 |
| F | ✓ | ✗ | ✓ | ✗ | ✗ | Visual | 0.9306 | 0.9261 | 0.9313 | 0.9120 | 0.9167 | 0.9295 | 0.8928 | 0.8696 | 0.9169 |
|  |  |  |  |  |  | Audio | 0.8295 | 0.8128 | 0.8389 | 0.7274 | 0.6984 | 0.7194 | 0.8693 | 0.8723 | 0.8685 |
|  |  |  |  |  |  | Fused | 0.9063 | 0.9091 | 0.9027 | 0.8462 | 0.8696 | 0.8517 | 0.8432 | 0.8240 | 0.8387 |
| F1 | ✓ | ✗ | ✓ | ✓ | ✗ | Fused | 0.9120 | 0.9164 | 0.9200 | 0.8523 | 0.8800 | 0.8656 | 0.8514 | 0.8333 | 0.8459 |
| F2 | ✓ | ✗ | ✓ | ✗ | ✓ | Fused | 0.9200 | 0.9184 | 0.9120 | 0.8571 | 0.8715 | 0.8609 | 0.8499 | 0.8352 | 0.8420 |
| F3 | ✓ | ✗ | ✓ | ✓ | ✓ | Fused | 0.9286 | 0.9261 | 0.9259 | 0.8696 | 0.8854 | 0.8750 | 0.8528 | 0.8415 | 0.8524 |
| G | ✗ | ✓ | ✓ | ✗ | ✗ | Visual | 0.9184 | 0.9063 | 0.9024 | 0.8977 | 0.8799 | 0.8864 | 0.8750 | 0.8523 | 0.8715 |
|  |  |  |  |  |  | Audio | 0.8182 | 0.8080 | 0.8295 | 0.7135 | 0.6923 | 0.7309 | 0.8696 | 0.8683 | 0.8715 |
|  |  |  |  |  |  | Fused | 0.8854 | 0.8715 | 0.8661 | 0.8438 | 0.8399 | 0.8333 | 0.8588 | 0.8333 | 0.8696 |
| G1 | ✗ | ✓ | ✓ | ✓ | ✗ | Fused | 0.8926 | 0.8799 | 0.8799 | 0.8588 | 0.8455 | 0.8409 | 0.8594 | 0.8409 | 0.8770 |
| G2 | ✗ | ✓ | ✓ | ✗ | ✓ | Fused | 0.8932 | 0.8848 | 0.8736 | 0.8522 | 0.8462 | 0.8455 | 0.8529 | 0.8399 | 0.8727 |
| G3 | ✗ | ✓ | ✓ | ✓ | ✓ | Fused | 0.9063 | 0.8928 | 0.8912 | 0.8696 | 0.8621 | 0.8594 | 0.8696 | 0.8523 | 0.8973 |
| H | ✓ | ✓ | ✓ | ✗ | ✗ | Visual | 0.9600 | 0.9600 | 0.9948 | 0.9643 | 0.9649 | 0.9692 | 0.9333 | 0.9111 | 0.9551 |
|  |  |  |  |  |  | Audio | 0.8571 | 0.8333 | 0.8715 | 0.7463 | 0.7143 | 0.7659 | 0.9310 | 0.9111 | 0.9131 |
|  |  |  |  |  |  | Fused | 0.9375 | 0.9312 | 0.9248 | 0.9091 | 0.8958 | 0.9183 | 0.8621 | 0.8444 | 0.8919 |
| H1 | ✓ | ✓ | ✓ | ✓ | ✗ | Fused | 0.9479 | 0.9477 | 0.9419 | 0.9231 | 0.9167 | 0.9401 | 0.8909 | 0.8710 | 0.9037 |
| H2 | ✓ | ✓ | ✓ | ✗ | ✓ | Fused | 0.9408 | 0.9383 | 0.9455 | 0.9286 | 0.9200 | 0.9312 | 0.8814 | 0.8667 | 0.9146 |

## G.3 NOTATIONS IN OUR METHOD

Table 10 lists all notations that appear in our method and their corresponding descriptions.

# H MORE CASE STUDY

## H.1 CASE STUDY OF TDAM

Before assessing the effectiveness of TDAM, it is helpful to recall a basic premise of deception detection: diagnostic cues rarely appear as isolated static expressions. Instead, they unfold as temporal micro-changes-brief activations of facial muscles, blinks or pupillary dilation/constriction, and rhythmic head movements. These cues are inherently time-dependent. However, conventional pipelines often decompose a video into individual frames and encode them independently, a practice

Table 10: Notations and their corresponding descriptions used in SPOT-JS.

| Notations | Description | Notations | Description |
|---|---|---|---|
| $\mathbb{R}$ | Real number space | $\mathbb{C}$ | Complex domain for spectra |
| $t \in \{a, v\}$ | Modality indicator: audio ($a$) / video ($v$) | $y \in \{0, 1\}$ | Class label (1 deceptive, 0 truthful) |
| $\mathbf{D} = \{\mathcal{X}, \mathcal{Y}\}$ | Multimodal dataset | $\hat{y}$ | Predicted label |
| $x = \{x^a, x^v\}$ | A multimodal input pair | $f : \mathcal{X} \to \mathcal{Y}$ | Decision function |
| $x_v, x_a$ | Raw video / audio signals | $\hat{x}_v, \hat{x}_a$ | Normalized/resampled inputs after TDAM |
| $T, \tau_0, \tau$ | Video duration, start time, timestamp | $N$ | Number of uniformly sampled key frames |
| $f_i$ | $i$-th sampled frame | $\phi(\cdot) / T(\cdot)$ | Color conversion / normalization transforms |
| $B, H, W$ | Batch size, frame height, frame width | $f_s, f'_s$ | Original / adjusted audio sampling rate |
| $x_t$ | Encoded sequence for modality $t$ | `VideoMAE`/ `W2V2` | Visual / audio encoders |
| $\mathbf{X}^t[k]$ | DFT of $x_t$ at frequency index $k$ | $F_{\text{seq}}/F_{\text{seq}}^{-1}$ | 1D DFT / IDFT along sequence dim. |
| $\mathbf{X}^t \in \mathbb{C}^{B \times N \times D}$ | Spectrum tensor (batch $\times$ length $\times$ dim) | $D$ | Feature dimension |
| $j$ | Imaginary unit in DFT ($e^{-j\cdot}$) | $\|\mathbf{X}^t\|^2$ | Power spectrum of modality $t$ |
| $\mathbf{C}^t = [C_1^t, \dots, C_k^t]$ | Learnable Chebyshev coefficients | $\mathbf{K}^t = [k_1^t, \dots, k_k^t]$ | Filter bank in LCSF |
| $\alpha, \theta_{\text{base}}$ | Learnable factor and base angle in LCSF | $\odot$ | Element-wise multiplication |
| $\tilde{\mathbf{X}}_v, \tilde{\mathbf{X}}_a$ | Projected spectra for video/audio | $\mathbf{W}_s, \mathbf{W}_t \in \mathbb{R}^{D \times D}$ | Linear projections for OTCF |
| $\mathbf{M}$ | Ground cost matrix (cosine distance $1 - \cos$) | $\mathbf{T}$ | Entropic-regularized OT transport plan |
| $\langle \cdot, \cdot \rangle_F$ | Frobenius inner product | $\Pi(p, q)$ | Set of couplings with marginals $p, q$ |
| $\epsilon, \mathcal{H}(\cdot)$ | Entropic reg. weight and entropy | $z_t$ | Residually fused spectral feat. in OTCF |
| $Z_t$ | $z_t$ mapped back to spatial domain via IDFT | $J$ | JS divergence between $Z_v$ and $Z_a$ |
| $\text{KL}(\cdot\|\cdot)$ | Kullback-Leibler divergence | $\mathbf{M}_{JS} = \frac{1}{2}(Z_v + Z_a)$ | Mean distribution used in JS-Align |
| $\mathbf{W}_a, \mathbf{W}_v$ | Trainable fusion weights | $\mathbf{f}$ | Final fused representation after JS-Align |
| $\mathcal{F}_1, \mathcal{F}_2, \mathcal{F}_3$ | MLPs for mapping / transfer / classification | $\mathbf{f}' = \mathcal{F}_1(\mathbf{f})$ | Mapped target features ($\mathbb{R}^{n \times d}$) |
| $n, d$ | Batch size and feature dim in CSKT | $L_s$ | # classes of source VFER dataset (e.g., DFEW) |
| $\mathcal{Q}, \mathcal{Q}^k$ | Uniform dist. over $L_s$ classes / prototype of class $k$ | $\mathcal{P}$ | Uniform dist. over $n$ target samples |
| $\delta_{(\cdot)}$ | Dirac measure at a point | $J_k$ | # samples in source class $k$ |
| $\mathbf{X}_j^{s,k}$ | $j$-th source sample in class $k$ (VFER) | $p_j^k$ | Importance of sample $j$ in class $k$ |
| $\mathbf{T}^{\text{low},k}$ | Low-level OT plan (mini-batch $\leftrightarrow$ class-$k$ samples) | $\mathbf{M}^{\text{low},k}$ | Low-level cost (cosine distance) |
| $\mathbf{X}^{\text{trans}}$ | Transferred knowledge from source domain | $\mathbf{X}^{\text{fused}}$ | $\xi' \mathbf{X}^{\text{trans}} + (1 - \xi')\mathbf{f}'$ |
| $\xi, \xi'$ | Max / current weight of transferred features | $e, N_e$ | Current / total epochs (curriculum for $\xi'$) |
| $L_t$ | # classes in target deception task | $\hat{y}$ | Classifier output in $\mathbb{R}^{n \times L_t}$ |
| $\mathcal{L}_{ce}$ | Cross-entropy loss | $\mathcal{L}_{ot}, \eta$ | Sinkhorn divergence term and its weight |
| $\Delta_n$ | Probability simplex in $\mathbb{R}^n$ | $\mathbb{E}[\cdot]$ | Expectation operator |

that conflicts with established observational principles in deception research and risks missing cross-frame dynamics and causal ordering. In contrast, our module adheres to this temporal nature by operating on contiguous frame sequences, enabling the model to observe how cues evolve over time. This design choice underpins the effectiveness of the TDAM.

## H.2 FURTHER VISUALIZATION STUDY

In this section, we conduct further visual analysis to investigate the generalization capability and effectiveness of the proposed model.

In the main text, we explored scenarios where the model was trained and tested on a single dataset. To further evaluate its performance, we visualize the results of the model trained on the RLT dataset and tested on the DOLOS dataset. The RLT dataset contains high-risk courtroom deception scenarios, whereas DOLOS comprises low-risk deception scenarios from a large-scale gaming environment. As shown in Fig. 5, even when evaluated across different domains, the model consistently identifies

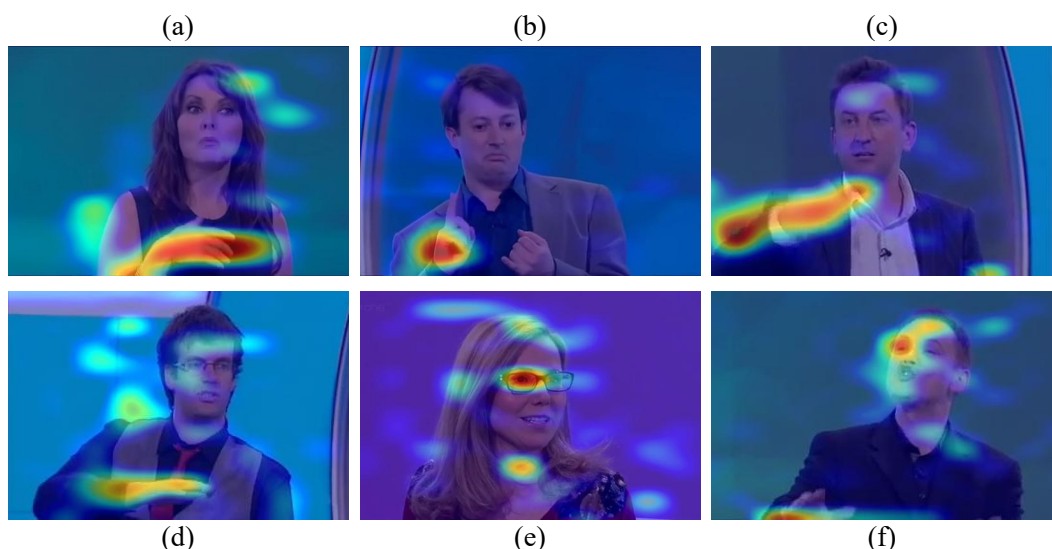

(a)  (b)  (c)

(d)  (e)  (f)

Figure 5: More visualization cases. We visualize the cases trained on the RLT dataset and tested on the DOLOS dataset.

subtle key features. Subfigures (a) to (d) and (f) clearly demonstrate the models sustained attention to hand gestures, which have been established in traditional deception detection as indicators of lying. Subfigures (e) and (f) reveal that the model continues to focus on the eye region. Physiological responses such as pupil dilation and blink rate are well documented cues in conventional deception research and serve as critical indicators for deceit. These examples illustrate that our model successfully attends to various deception clues that are empirically validated in traditional studies, even when deployed in divergent real-world settings. Together with the case studies presented in the main text, these visual analyses elucidate the reasons behind our model's strong performance in both within-domain and cross-domain deception detection, underscoring the superiority of our framework.

By our LCSF

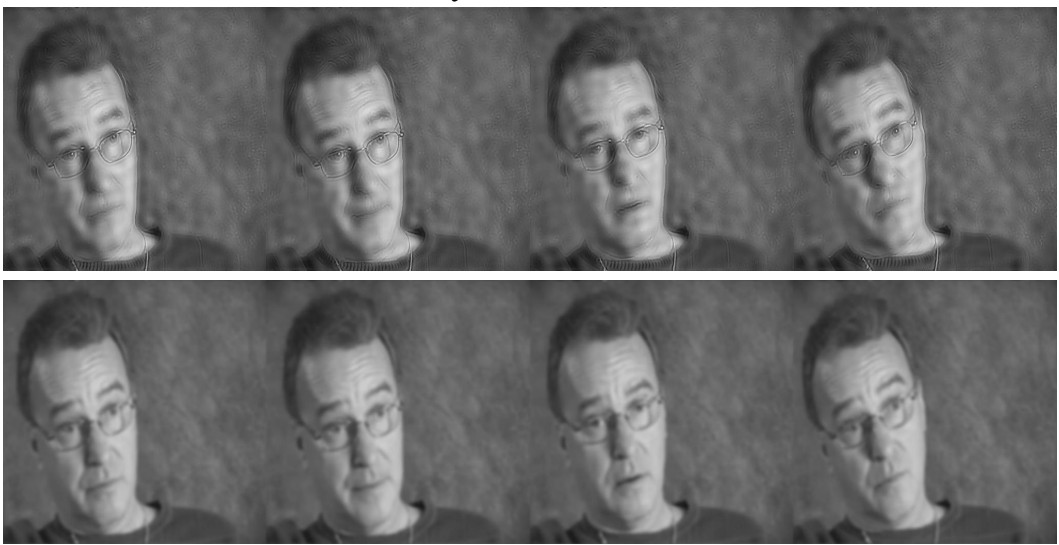

By  Gaussian filter

Figure 6: Visualization diagrams for detailed analysis after LCSF module filtering. The upper section shows the images processed by our LCSF module, while the lower section displays the results obtained using Gaussian filtering.

## H.3 CASE STUDY OF LCSF MODULE

To clarify the function of the LCSF module, we visualized its filtered outputs, as shown in Fig. 6. After processing through this module, facial details closely associated with deceptive behaviorincluding pupil dynamics and blink patterns in the eye region, subtle facial muscle movements, skin texture variations, and lip morphologybecome more distinct, with noticeably enhanced contrast. These enhanced features provide critical visual evidence for deception identification. Additionally, low-frequency behavioral patterns such as head movements are effectively extracted within the frequency domain, clearly distinguishable from the background. This observation aligns with our earlier visualizations showing the models focus on the head region. In comparison, although traditional Gaussian filtering can suppress image noise and preserve low-frequency information, it tends to blur the image overall and fails to retain discriminative subtle behavioral cues. The output lacks sufficient detail in key regions such as the eyes, muscle movements, skin folds, and the mouth, making it difficult to support physiological behavior-based deception analysis. This comparison further validates the effectiveness and specificity of the LCSF module in enhancing discriminative frequency components.

## I ADDITIONAL EXPERIMENTS ON LCSF

Table 11: Performance comparison on MIntRec and MIntRec2.0 datasets.

| Methods | MIntRec | | | | MIntRec 2.0 | | | |
|---|---|---|---|---|---|---|---|---|
| | ACC (%) | WF1 (%) | WP (%) | R (%) | ACC (%) | WF1 (%) | WP (%) | R (%) |
| MulT (Tsai et al., 2019) | 72.52 | 71.80 | 72.60 | 67.44 | 56.95 | 54.26 | 54.49 | 40.65 |
| MAG-BERT (Hasan et al., 2020) | 72.16 | 71.30 | 72.03 | 67.61 | 55.87 | 52.58 | 53.71 | 39.93 |
| TCL-MAP (Zhou et al., 2024) | 73.69 | 73.38 | 73.90 | 71.59 | 56.99 | 54.33 | 55.07 | 41.87 |
| MVCL-DAF (Hu et al., 2025a) | 74.72 | 74.61 | 75.07 | 71.94 | 57.80 | 55.05 | 55.82 | 42.03 |
| **MVCL-DAF+LCSF (Ours)** | **76.40** | **76.33** | **76.67** | **74.35** | **60.26** | **59.19** | **60.54** | **54.12** |
| **Performance Improvement** | **+1.68** | **+1.72** | **+1.60** | **+2.41** | **+2.46** | **+4.14** | **+4.72** | **+12.09** |

To validate the effectiveness and generalization capability of the LCSF module, we conducted additional experiments in the field of intent recognition. As shown in Table 11, the integration of our proposed LCSF module brings significant performance improvements to the MVCL-DAF model,demonstrating the strong generalization and effectiveness of our approachOn the MintRec (Zhang et al., 2022) dataset, our enhanced method demonstrates remarkable gains compared to the baseline without LCSF: Accuracy (ACC) improves by 2.36%, Weighted F1-score (WF1) increases by 2.27%, Weighted Precision (WP) rises by 2.73%, and Recall (R) enhances by 3.69%. Similarly, evaluations on the MintRec 2.0 (Zhang et al., 2024) multi-turn emotional dialogue dataset show substantial performance improvements: ACC increases by 2.60%, WF1 by 4.30%, WP by 4.81%, and R by 12.09%.

## J ADDITIONAL EXPERIMENTS BASED ON ATTENTION

Table 12: Results with fused modalities. Comparison of our method with attention-based fusion methods.

| Target | RLT | | | DOLOS | | | BOL | | |
|---|---|---|---|---|---|---|---|---|---|
| Method | F1 | ACC | AUC | F1 | ACC | AUC | F1 | ACC | AUC |
| Attention | 0.9432 | 0.9375 | 0.9410 | 0.9205 | 0.9145 | 0.9376 | 0.8864 | 0.8656 | 0.9116 |
| SPOT-JS(Ours) | **0.9630** | **0.9600** | **0.9679** | **0.9474** | **0.9474** | **0.9846** | **0.9153** | **0.8889** | **0.9393** |
| | +1.98% | +2.25% | +2.69% | +2.69% | +3.29% | +4.70% | +2.89% | +2.33% | +2.77% |

To provide an intuitive comparison with attention-based fusion methods, we conducted comparative experiments. As shown in Table 12, our fusion method demonstrates superior performance compared to attention-based fusion approaches.

# K   PSEUDO-CODE OF SPOT-JS

The pseudo-code of SPOT-JS in training phase and testing phase are shown in Algorithm 1 and Algorithm 2, respectively.

---

**Algorithm 1** Training Algorithm of SPOT-JS

---

**Input:** Raw video-audio sample $\mathbf{V}$ with synchronized streams; Source VFER knowledge $\mathbf{X}^{s_0}$ (DFEW); model params $\Theta$
**Output:** Updated params $\Theta$; prediction $\hat{\mathbf{y}}$

1: *# One iteration (minibatch)*
2: *§ TDAM: Preprocess & Temporal Alignment*
3: Sample frames and normalize: $\hat{\mathbf{x}}^v = T\big(\{\varphi(f_i)\}_{i=1}^N\big)$    *# Eq. (1),(2)*
4: Extract & resample audio aligned to video: $\hat{\mathbf{x}}^a = R(A(\mathbf{V}), f_s')$    *# Eq. (3)*
5: *§ Feature Encoding*
6: $\mathbf{x}^v = \text{VideoMAE}(\hat{\mathbf{x}}^v);$    $\mathbf{x}^a = \text{W2V2}(\hat{\mathbf{x}}^a)$    *# Eq. (4),(5)*
7: *§ DFT to Frequency Domain*
8: $\mathbf{X}^t = \mathcal{F}_{seq}(\mathbf{x}^t)$ for $t \in \{a, v\}$    *# Eq. (6)*
9: *§ LCSF: Learnable Chebyshev Spectrum Filter* ★
10: Compute power spectra and filter: $\hat{\mathbf{X}}^t = \sum_{i=1}^k |\mathbf{X}^t|^2 \odot \mathbf{k}_i^t C_i^t;$   $C_i^t = \cos((2i-1)\theta_{base})$ *# Eq.* (7),(8)
11: *§ OTCF: Entropic-OT Cross-Modal Fusion (bi-directional)*
12: **for** $(s, t) \in \{(v, a), (a, v)\}$ **do**
13:     Project to shared space: $\tilde{\mathbf{X}}^s = \hat{\mathbf{X}}^s \mathbf{W}_s, \tilde{\mathbf{X}}^t = \hat{\mathbf{X}}^t \mathbf{W}_t$ *# Eq. (11)*
14:     Cost by cosine distance: $\mathbf{M} = 1 - \cos(\tilde{\mathbf{X}}^s, \tilde{\mathbf{X}}^t)$ *# Eq. (12)*
15:     Sinkhorn to solve entropic OT, get plan $\mathbf{T}$ *# Eq. (10)*
16:     Residual transport: $\mathbf{z}_s = \mathbf{T}\tilde{\mathbf{X}}^t + \tilde{\mathbf{X}}^s$ *# Eq. (13)*
17:     Back to spatial: $\mathbf{Z}_s = \mathcal{F}_{seq}^{-1}(\mathbf{z}_s)$ *# Eq. (14)*
18: **end for**
19: *§ JS-Align: Jensen-Shannon Guided Alignment*
20: $J = \text{JS}(\mathbf{Z}_v \| \mathbf{Z}_a);$    $\mathbf{f} = (1-J)(\mathbf{W}_a\mathbf{Z}_a + \mathbf{W}_v\mathbf{Z}_v) + J\mathbf{Z}_a + J\mathbf{Z}_v$ *# Eq. (15),(16)*
21: *§ CSKT: Chebyshev Spectrum-guided Knowledge Transfer* ★
22: Filter source knowledge: $\mathbf{X}^s = \text{LCSF}(\mathcal{F}_{seq}(\mathbf{X}^{s_0}))$ *# Appx Eq. (1)*
23: Map target: $\mathbf{f}' = \mathcal{F}_1(\mathbf{f});$ build $P = \frac{1}{n}\sum_i \delta_{\mathbf{f}_i'}, Q = \frac{1}{L_s}\sum_k \delta_{\mathbf{Q}_k}$
24: **for** $k = 1$ **to** $L_s$ **do**
        *# low-level OT to class-$k$*
25:     $\mathbf{M}_{i,j}^{low,k} = 1 - \cos(\mathbf{f}_i', \mathbf{X}_j^{s,k})$
26:     Solve $\mathbf{T}^{low,k}$ by Sinkhorn *# Appx Eq. (4)*
27:     $\mathbf{M}_{:,k} = \langle \mathbf{T}^{low,k}, \mathbf{M}^{low,k} \rangle_F$ *# as cost for high-level*
28: **end for**
29: High-level OT: $\mathbf{T} = \arg\min_{T \in \Pi(P,Q)} \langle T, \mathbf{M} \rangle_F - \epsilon H(T)$ *# Appx Eq. (2)*
30: Transfer & IDFT: $\mathbf{X}^{trans} = \mathcal{F}_2\big(n\sum_k T_{:,k} \overline{\mathbf{X}}^{s,k}\big);$   $\mathbf{X}^{trans} \leftarrow \mathcal{F}_{seq}^{-1}(\mathbf{X}^{trans})$ *# Appx Eq.* (5),(6)
31: Curriculum fuse: $\mathbf{X}^{fused} = \xi' \mathbf{X}^{trans} + (1-\xi')\mathbf{f}'$ *# Appx Eq. (7)*
32: *§ SRKB bank update (adopted)* ★
33: Update knowledge bank $\mathbf{B}$ (sample-specific re-weighting)
34: *§ Classification, Loss, Backward*
35: $\hat{\mathbf{y}} = \mathcal{F}_3(\mathbf{X}^{fused})$ *# Appx Eq. (8)*
36: $\mathcal{L}_{ce} = -\mathbb{E}_y[\log \hat{\mathbf{y}}];$    $\mathcal{L}_{ot} = d^{sOT}(P,Q) - \frac{1}{2}d^{sOT}(P,P) - \frac{1}{2}d^{sOT}(Q,Q)$ *# Appx Eq. (9)-(10)*
37: $\mathcal{L} = \mathcal{L}_{ce} + \eta \mathcal{L}_{ot};$    **Backward**$(\mathcal{L})$ *# Appx Eq. (11)*

---

---

**Algorithm 2** Testing Algorithm of SPOT-JS

---

**Input:** Raw video-audio sample $\mathbf{V}$; frozen banks/backbones $\Theta$
**Output:** Predicted label $\hat{y}$
1: # *One iteration (inference)*
2: § *TDAM*     # Eq. (1)-(3)
3: § *Feature Encoding*     # Eq. (4),(5)
4: § *DFT*     # Eq. (6)
5: § *LCSF*     # Eq. (7),(8)
6: § *OTCF (bi-dir) + IDFT*     # Eq. (11)-(14)
7: § *JS-Align to get* $\mathbf{f}$     # Eq. (15),(16)
8: § *CSKT transform & fuse (frozen SRKB)*     # Appx Eq. (1)-(7)
9: § *Classifier*     $\hat{y} = \mathcal{F}_3(\mathbf{X}^{fused})$ # Appx Eq. (8)

---

# L CODE

Our codes is available: `https://anonymous.4open.science/r/9BFE/`

