# OpenReview forum: "SPOT-JS:Spectral Chebyshev Filter and Optimal Transport Fusion with Jensen-Shannon Alignment for Cross-Domain Multimodal Deception Detection"
_ICLR.cc/2026/Conference — ICLR 2026 Conference Withdrawn Submission_

### Official Review · Reviewer_18Hc · 2025-10-24

**Soundness:** 3
**Presentation:** 3
**Contribution:** 3
**Rating:** 6
**Confidence:** 4

**Summary:**

This paper proposes SPOT-JS, a novel framework for multimodal deception detection, introducing several key modules to enhance the generalization of the model and the effectiveness of multimodal fusion. Specifically, TDAM enhances model generalization through standardized data preprocessing, while the CSKT module utilizes spectral filtering to improve knowledge transfer from facial expression datasets. Besides, following modules are introduced  to enhance the effectiveness of multimodal fusion: 1) OTCF, a fusion module based on optimal transmission, to find fine-grained correspondence between audo and video spectra; 2) JS Align module, using JS divergence to adaptively weight the fused representation; In addition, the author also proposed a learnable Chebyshev spectral filter (LCSF) that enhances single modal features by focusing on frequency bands relevant to the task. The author conducted extensive experiments on three benchmark datasets, and the results showed that the method performed significantly better than many baseline methods within the domain, especially in cross domain settings.

**Strengths:**

This method innovatively shifts feature enhancement and multimodal fusion to the frequency domain for processing, introducing theories such as Optimal Transport (OT) and JS divergence to guide alignment, which significantly improves the performance of the deception detection model. Furthermore, through operations like data preprocessing and feature filtering, it demonstrates outstanding performance in addressing the critical cross-domain generalization problem.

**Weaknesses:**

1.The author points out that multimodal feature fusion based on attention mechanisms is relatively coarse and thus yields suboptimal performance. Consequently, they propose a fusion method based on optimal transport theory. Can the differences in the feature fusion outcomes between these two approaches be demonstrated intuitively or measured quantitatively?

2.While the paper provides some visualizations, the interpretability of the frequency-domain operations remains somewhat abstract. For instance, what specific frequency bands does the LCSF learn to amplify for deception? Correlating these learned filters back to known psycho-physiological cues (e.g., high-frequency voice tremors, low-frequency head movements) would provide invaluable insights.

**Questions:**

See weakness

---

> ### Author Response · Authors · 2025-11-19
> **(1)Response to Reviewer 18Hc weak1**
>
> We are truly honored to have received your valuable professional feedback. Your insights have been immensely beneficial, and we appreciate your scoring.
> Now, please allow me to address your questions with the following responses:
> (1)Regarding the first question, we appreciate your comment. We acknowledge our oversight in not including intuitive results for demonstration. To address this, we have conducted additional experiments to illustrate the performance differences. Detailed experimental results can be found in Section J on page 26 of the paper. As demonstrated in these experiments, our method shows clear advantages compared to attention-based fusion approaches.
>
> *Table: Results with fused modalities. Comparison of our method with attention-based fusion methods.*
> | **Target** | **RLT** | | | **DOLOS** | | | **BOL** | | |
> |------------|---------|---------|---------|-----------|---------|---------|---------|---------|---------|
> | **Method** | **F1** | **ACC** | **AUC** | **F1** | **ACC** | **AUC** | **F1** | **ACC** | **AUC** |
> | Attention | 0.9432| 0.9375 | 0.9410 | 0.9205 | 0.9145 | 0.9376 | 0.8864 | 0.8656 | 0.9116 |
> | SPOT-JS (Ours) | **0.9630** | **0.9600** | **0.9679** | **0.9474** | **0.9474** | **0.9846** | **0.9153** | **0.8889** | **0.9393** |
> | | **+1.98%** | **+2.25%** | **+2.69%** | **+2.69%** | **+3.29%** | **+4.70%** | **+2.89%** | **+2.33%** | **+2.77%** |

---

> ### Author Response · Authors · 2025-11-19
> **(2)Response to Reviewer 18Hc weak2**
>
> (2)Regarding the second question, we sincerely appreciate your insightful comment. While visualizations were included in the original manuscript, we acknowledge that they lacked professional interpretation regarding specific frequency bands and related cues as you pointed out. To provide an intuitive demonstration of the enhanced frequency bands, we have incorporated filter visualizations of our LCSF module and compared them with commonly used Gaussian filtering in the appendix (Figure 6, page 25).
>
> As shown in the visualizations, our LCSF module significantly amplifies facial details while effectively blurring irrelevant regions. Upon closer examination of the magnified images, it becomes evident that the eye regions processed by our LCSF module remain clearly visible with enhanced brightness and contrast. This provides valuable cues for deception detection that relies on physiological indicators such as pupil variations and blink frequency. Furthermore, facial muscle movements, skin texture variations, and lip movements become more prominent and distinguishable, offering precious deception-related information.
>
> Specifically addressing your mention of low-frequency head movement cues, our LCSF module successfully distinguishes head regions from the surrounding environment. We believe our module effectively extracts relevant frequency bands, which is further corroborated by our earlier visualizations showing the model's detailed attention to head regions. This comprehensive analysis adequately addresses your concerns. We also recognize that extracting specific physiological features prior to applying the LCSF module might yield improved results, an aspect we plan to investigate in future work.
>
> In contrast, images processed with conventional Gaussian filtering appear uniformly blurred and fail to highlight any useful deception cues. Fundamentally, Gaussian blur operates by replacing pixel values with weighted averages of neighboring pixels, which while effective for suppressing Gaussian noise during digital signal conversion, proves inadequate for identifying subtle physiological signals. Although it preserves low-frequency components, Gaussian filtering provides limited assistance for deception clue recognition. The magnified examples demonstrate that while Gaussian-filtered images maintain overall recognizability, they lack detailed deception-related information: eyes appear blurred and difficult to observe, facial muscles and skin textures become obscured, and lip movements are indistinguishable. Regarding head movement analysis, Gaussian processing blends subjects seamlessly with their background environment, offering no distinctive features for deception identification.
>
> To further validate LCSF's effectiveness and generalization capability, we extended its application to intent recognition tasks. Please refer to Section I on page 26, where we integrated our module with the latest SOTA method from AAAI 2025. The results demonstrate substantial performance improvements across both datasets, with advancement margins significantly exceeding those reported in the original SOTA publication. This strongly confirms LCSF's effectiveness as a plug-and-play solution.
>
> In summary, we believe our LCSF module successfully addresses your concerns regarding physiological cue extraction while demonstrating excellent generalization capability. We hope this response adequately resolves your questions, and we thank you again for your professional insights, which have significantly enhanced our paper's completeness and interpretability. We would be pleased to address any further questions you might have and look forward to your response.
>
>
> # Performance comparison on MIntRec and MIntRec2.0 datasets.
>
> | Methods | MIntRec ACC (%) | MIntRec WF1 (%) | MIntRec WP (%) | MIntRec R (%) | MIntRec 2.0 ACC (%) | MIntRec 2.0 WF1 (%) | MIntRec 2.0 WP (%) | MIntRec 2.0 R (%) |
> | :--- | :--- | :--- | :--- | :--- | :--- | :--- | :--- | :--- |
> | MulT  | 72.52 | 71.80 | 72.60 | 67.44 | 56.95 | 54.26 | 54.49 | 40.65 |
> | MAG-BERT  | 72.16 | 71.30 | 72.03 | 67.61 | 55.87 | 52.58 | 53.71 | 39.93 |
> | TCL-MAP  | 73.69 | 73.38 | 73.90 | 71.59 | 56.99 | 54.33 | 55.07 | 41.87 |
> | MVCL-DAF [2025AAAI][1] | 74.72 | 74.61 | 75.07 | 71.94 | 57.80 | 55.05 | 55.82 | 42.03 |
> | **MVCL-DAF+LCSF (Ours)** | **76.40** | **76.33** | **76.67** | **74.35** | **60.26** | **59.19** | **60.54** | **54.12** |
> | **Performance Improvement** | +1.68 | +1.72 | +1.60 | +2.41 | +2.46 | +4.14 | +4.72 | +12.09 |
>
>
> [1] Hu, B., Zhang, K., Zhang, Y., & Ye, Y. (2025, April). Adaptive Multimodal Fusion: Dynamic Attention Allocation for Intent Recognition. In Proceedings of the AAAI Conference on Artificial Intelligence (Vol. 39, No. 16, pp. 17267-17275).

---

> > ### Comment · Reviewer_18Hc · 2025-11-25
> >
> > Thanks for the response, which has solved my partial concerns. I will keep the rating for my side.

---

> > > ### Author Response · Authors · 2025-11-26
> > > **Response to Reviewer 18Hc**
> > >
> > > Thank you for your response, and we appreciate your valuable feedback once again.

---

### Official Review · Reviewer_7UMw · 2025-11-01

**Soundness:** 2
**Presentation:** 1
**Contribution:** 2
**Rating:** 2
**Confidence:** 2

**Summary:**

This work study the problem of multimodal (video-audio) deception detection, where the communicator is classified if they are performing the act of deception. The paper proposed a method SPOT-JS, which consists of a temporal deception alignment module (which is preprocessing module for both modalities), a learnable chebyshev spectrum filter that operates on the frequency space, an optimal transport cross-modal fusion, a Jensen-shannon guided alignment component, and a chebyshev spectrum guided knowledge transfer module. The proposed method is applied on three standard deception benchmarks, and achieved high classification performance.

**Strengths:**

- Detailed ablation experiments to validate the effectiveness of each components.
- Interesting case studies to show the effectiveness of the method.

**Weaknesses:**

- The paper proposed many technical components, but none of them is carefully studied or explained. There is no theoretical analysis or detailed exploration of the mechanism. Thus, it is unclear to me how each component of the proposed method is important, or how can they be applied on other classification or multimodal tasks.

- Readability of the paper is bad. The paper defines a lot of new jargons, e.g. the temporal deception alignment module (TDAM) essentially is just video and audio preprocessing steps. Also, key components of the method, e.g. loss function, is not included inside the main text but in appendix instead. The main text should be self-sustained.

- The literatures and benchmarks referenced seem to be out-of-dated. For literature review, especially section 2.2, there are a lot more recent works in frequency domain learning. Specifically, [1] seems to be highly relevant and similar to the proposed work. For benchmarks, [2] reported >99% and >99% accuracy and F1 score on RLT and DOLOS datasets using multiple baselines and methods, which are higher than what is reported in this paper. I am not extremely familiar with this subfield, can the authors explain the difference between the performance across baselines, and explain the performance difference?

[1] Lao, A., Zhang, Q., Shi, C., Cao, L., Yi, K., Hu, L., & Miao, D. (2024, March). Frequency spectrum is more effective for multimodal representation and fusion: A multimodal spectrum rumor detector. In Proceedings of the AAAI conference on artificial intelligence (Vol. 38, No. 16, pp. 18426-18434).

[2] Zhuo, Y., Baskaran, V. M., Wang, L. Y. K., & Phan, R. C. W. (2025). @ LM DeceptionNet: A multimodal approach for efficient transfer learning-based deception detection. Knowledge-Based Systems, 113499.

**Questions:**

NA

---

> ### Author Response · Authors · 2025-11-19
> **(1)Response to Reviewer 7UMw weak1**
>
> Thank you for your valuable feedback. Given that you may not be extensively familiar with this field, allow me to provide a detailed explanation:
> (1)Thank you for your first question. I sincerely apologize that due to page limitations, the description of our method in the paper had to be kept concise, which may have caused some confusion. Let me provide further clarification with additional examples and experimental insights.
>
> Regarding the TDAM module, we believe it can be applied to any video-based classification task or datasets containing video sequences. In many existing approaches, features are pre-extracted using models such as ResNet-50 and then utilized in downstream tasks. However, such frame-level features may not always be sufficiently discriminative. In such cases, our proposed TDAM module can be employed to extract more informative and temporally coherent representations.
>
> As for the LCSF module, it is designed as a plug-and-play component. To offer a more intuitive understanding, we applied it to the field of multimodal intent recognition, where it also demonstrated promising performance. As shown in the provided table, the module brings clear improvements. Furthermore, we conducted visual filtering experiments and compared it with the widely-used Gaussian filtering. Please refer to Figure 6 on page 25 of the appendix. It can be observed that our LCSF module effectively enhances facial details while suppressing irrelevant regions. For example, upon closer inspection, the eye region processed by LCSF remains clearly visible, with improved brightness and contrast—providing valuable cues for deception detection that often relies on pupil dynamics and blink frequency. Similarly, facial muscle movements, skin textures, and lip variations are also more pronounced, offering critical insights for detecting deceptive behavior.
>
> In contrast, the Gaussian-filtered version appears generally recognizable but fails to highlight meaningful deception-related details. When zooming in, the eyes become blurred and hard to discern, and facial muscle movements or lip changes are no longer clearly observable. This limits its utility in providing actionable clues for deception detection.
>
> Regarding its application, the LCSF module can be applied after feature extraction or at other stages of the pipeline. In our current implementation, it is applied to individual modality features, but it could also be used on fused features or inspire other novel extensions.
>
> As for the OTCF and JS-Align modules, you may follow the fusion strategies described in the paper—such as integrating two different modalities, or, in the case of three modalities, performing pairwise fusion. Moreover, more flexible usage patterns can be explored, such as refining OTCF with more adaptive optimal transport formulations or introducing learnable hyperparameters to enhance its flexibility and performance.
>
> # Performance comparison on MIntRec and MIntRec2.0 datasets.
> | Methods | MIntRec ACC (%) | MIntRec WF1 (%) | MIntRec WP (%) | MIntRec R (%) | MIntRec 2.0 ACC (%) | MIntRec 2.0 WF1 (%) | MIntRec 2.0 WP (%) | MIntRec 2.0 R (%) |
> | :--- | :--- | :--- | :--- | :--- | :--- | :--- | :--- | :--- |
> | MulT  | 72.52 | 71.80 | 72.60 | 67.44 | 56.95 | 54.26 | 54.49 | 40.65 |
> | MAG-BERT  | 72.16 | 71.30 | 72.03 | 67.61 | 55.87 | 52.58 | 53.71 | 39.93 |
> | TCL-MAP  | 73.69 | 73.38 | 73.90 | 71.59 | 56.99 | 54.33 | 55.07 | 41.87 |
> | MVCL-DAF [2025AAAI][1] | 74.72 | 74.61 | 75.07 | 71.94 | 57.80 | 55.05 | 55.82 | 42.03 |
> | **MVCL-DAF+LCSF (Ours)** | **76.40** | **76.33** | **76.67** | **74.35** | **60.26** | **59.19** | **60.54** | **54.12** |
> | **Performance Improvement** | +1.68 | +1.72 | +1.60 | +2.41 | +2.46 | +4.14 | +4.72 | +12.09 |
>
> [1] Hu, B., Zhang, K., Zhang, Y., & Ye, Y. (2025, April). Adaptive Multimodal Fusion: Dynamic Attention Allocation for Intent Recognition. In Proceedings of the AAAI Conference on Artificial Intelligence (Vol. 39, No. 16, pp. 17267-17275).

---

> ### Author Response · Authors · 2025-11-19
> **(2)Response to Reviewer 7UMw weak2**
>
> (2)Thank you for your second question and valuable feedback. We sincerely apologize that due to page limitations, the loss function was not included in the main text initially. We have now incorporated it into the main body of the paper as per your suggestion.
>
> You rightly pointed out that the description of the TDAM module in the paper was relatively brief, with only a concise mathematical formulation provided. Indeed, the TDAM module is a data processing component, and we gave it a distinctive name for clarity. We are open to revising the terminology if you have better suggestions. However, as the classic adage in deep learning goes, "garbage in, garbage out"—the quality of input data profoundly impacts model performance. Thus, improving how data is processed to enable models to capture more salient features is equally crucial.
>
> As noted in our paper, conventional data preprocessing methods often rely on isolated static frames or pre-processed individual images. However, deception detection requires the analysis of continuous movements—such as facial expressions, micro-gestures, and eye movements—which are inherently temporal in nature. To better capture these time-dependent cues, we proposed the TDAM module.
>
> To help illustrate the motivation behind our approach, we refer to a CVPR paper [3] as an example. To address asynchrony and temporal inconsistencies between facial expressions and body movements, the authors of [3] proposed a dual-stream network. One stream focuses on facial features, while the other aligns these features with five consecutive action frames. This idea of matching a static facial expression with a short sequence of motion frames led to notable improvements.
>
> In our work, we adopt a conceptually simple yet effective strategy: extracting consecutive frames as sequences and feeding them into the model to learn temporal deceptive patterns. While straightforward, experiments validate its effectiveness.
>
> To further clarify, consider the evolution from R-CNN to Fast R-CNN in object detection. Early approaches like R-CNN required extracting and processing thousands of region proposals per image—each independently—leading to high computational cost and significant redundancy due to overlapping regions. Fast R-CNN introduced a more efficient paradigm by extracting features from the entire image first, then performing region proposal, which markedly improved both speed and accuracy.
>
> Similarly, the core idea behind our method is simple and intuitive—yet experimentally validated as effective. We hope these explanations adequately address your concerns.

---

> ### Author Response · Authors · 2025-11-19
> **(3-1) Response to Reviewer 7UMw weak3**
>
> (3-1) Thank you for your third question regarding the literature and benchmark issues. First, I appreciate your correction on the frequency domain learning field. Due to time constraints and my focus on specific research directions, I have primarily read mathematical papers in the signal processing area, which resulted in my limited understanding of the frequency domain learning domain. Regarding the similarity issue, we conducted extensive attempts and tests to propose this module. We reviewed numerous mathematical literature and formulas for the design, including various Gaussian, Newton, and other formulations. Through various experiments, we utilized the fifth property of the first-kind Chebyshev polynomial. Additionally, we provided our code on page 28 of the paper. You can visit the website https://anonymous.4open.science/r/9BFE/README.md to view our CSKT.py code. From lines 31 to 159, you can see our designed LCSF module. It is evident that our code is significantly different from the code in the paper [1] you mentioned. Moreover, the formulas used in our code are distinct. In the paper you referenced, their code employs the formula:
> cos = torch.cos(torch.as_tensor((2 * (k + 1) - 1) * pi / 2 * self.num_filter))
>
> Whereas, as an example from lines 61-62 in our code:
> theta_base = (pi / (2.0 * K)) * F.softplus(alpha)
>
> theta = (2 * k - 1) * theta_base
>
> We can clearly observe that these are different. Firstly, the formulas themselves are distinct. The formula in the referenced code simplifies to (((2k+1)*pi)*K)/2, while our code uses the Chebyshev theorem formula ((2k-1)*pi)/(2K), which are entirely different. Furthermore, other aspects of the code design are also dissimilar. You can check lines 153 to 185 of their paper's code via the following link: https://github.com/dm4m/FSRU/blob/main/model.py.
>
> In terms of functionality, the paper you mentioned focuses on compressing the frequency spectrum, whereas our LCSF aims to highlight task-relevant frequency bands and suppress noise, which is fundamentally different. To demonstrate this disparity, we conducted visualizations after filtering, as described in our response (1). From the visualization images, you can observe that our LCSF module extracts more frequency bands related to deception clues.
>
> [1] Lao, A., Zhang, Q., Shi, C., Cao, L., Yi, K., Hu, L., & Miao, D. (2024, March). Frequency spectrum is more effective for multimodal representation and fusion: A multimodal spectrum rumor detector. In Proceedings of the AAAI conference on artificial intelligence (Vol. 38, No. 16, pp. 18426-18434).
>
> [2]Zhuo, Y., Baskaran, V. M., Wang, L. Y. K., & Phan, R. C. W. (2025). @ LM DeceptionNet: A multimodal approach for efficient transfer learning-based deception detection. Knowledge-Based Systems, 113499.
>
> [3]Ding, M., Zhao, A., Lu, Z., Xiang, T., & Wen, J. R. (2019). Face-focused cross-stream network for deception detection in videos. In Proceedings of the IEEE/CVF Conference on Computer Vision and Pattern Recognition (pp. 7802-7811).
>
> [4]Miah, M. M. M., Anika, A., Shi, X., & Huang, R. (2025). Hidden in Plain Sight: Evaluation of the Deception Detection Capabilities of LLMs in Multimodal Settings.  In Proceedings of the 63rd Annual Meeting of the Association for Computational Linguistics (Volume 1: Long Papers), pages 31013–31034, Vienna, Austria. Association for Computational Linguistics.

---

> ### Author Response · Authors · 2025-11-19
> **(3-2) Response to Reviewer 7UMw weak3**
>
> (3-2) Regarding the benchmark issues, we are pleased to provide the following clarifications. First, our paper employs the most recent benchmarks available. For instance, AFFAKT is from AAAI 2025, and AVA+CUFMCL is from 2024. Compared to Reference [2], our evaluation methodology differs. We use 5-fold cross-validation, which is the evaluation method adopted by the current state-of-the-art model, namely the AFFAKT paper (AAAI 2025) cited in our work. If a comparison with Reference [2] is insisted upon, it can only be roughly made with the 3-fold cross-validation results in their Table 5, as 3-fold cross-validation is approximately similar to 5-fold cross-validation. However, even this is not a precise comparison. Furthermore, the ACC for the DOLOS dataset under 3-fold cross-validation in their Table 5 does not reach the 99% you mentioned; the highest reported is only 80.02% ACC. You have evidently misinterpreted this.
>
> Secondly, the datasets used in Reference [2] appear to be incorrect. The DOLOS dataset contains only 1675 samples. Furthermore, since some videos in the dataset were banned on YouTube for various reasons, the actual usable dataset is likely even smaller. It is impossible for them to report 1678 samples, which exceeds the known total. This suggests that Reference [2] may have used additional external data, incorrect datasets, or other unstated issues, making the comparison unfair. Similarly, their use of the RLT dataset is questionable. The standard RLT dataset contains 121 complete videos; there are no known "corrupted videos" as claimed in Reference [2]. It appears they may have selectively removed some challenging samples to boost performance, which is unfair. Additionally, for the BOL dataset, they mention screening and removing data but fail to specify how many samples were deleted or the potential impact on performance. This raises significant concerns.
>
> Moreover, while the paper claims to propose a transfer learning method, it merely employs fine-tuning techniques, which does not constitute a genuine transfer learning approach as commonly understood. Given the above points, the comparability and reported performance metrics in Reference [2] are highly questionable. We remain skeptical of its validity and do not believe it is suitable for comparison.
>
> Taking the CVPR Reference [3] mentioned in our previous response (2) as an example, that paper uses a 10-fold cross-validation evaluation method. Under this method, they achieved 97.00% ACC and 99.78% AUC on the RLT dataset. However, this evaluation protocol is now considered outdated. The results in Table 4 of Reference [2] do not compare against any prior work nor specify their evaluation method clearly. We suspect they might be using an outdated protocol like 10-fold average, which could potentially explain their universally high results around 99.9%. Furthermore, we investigated a recent ACL 2025 paper [4], which employs a 4-fold cross-validation method (approximating 5-fold). Under this similar protocol, the best-reported ACC and AUC on the RLT dataset are only 83.47% and 83.44%, respectively.
>
> Regarding computational resources, you mentioned that Reference [2] requires high-configuration GPUs like the A100 (80GB). In contrast, our method runs efficiently on a single RTX 4090 (24GB) GPU. With a batch size of 16, it consumes only 16,861 MB of VRAM (approximately 1,053 MB per batch). This high efficiency demonstrates that our model can operate effectively in resource-constrained environments, which is a crucial advantage for the widespread deployment and practical application of deception detection systems.
>
> In summary, the most recent metrics from 2025 AAAI and 2025 ACL papers are also around 80%. The Reference [2] you cited clearly lacks the necessary conditions for a fair comparison. Importantly, our work represents a significant advancement over the current state-of-the-art (SOTA). Taking the video modality on the RLT dataset as an example, we achieved substantial improvements of +8.40% in F1, +9.30% in ACC, and +11.59% in AUC. Progress across all other datasets is similarly significant. We hope this explanation resolves your concerns and prevents any potential misunderstanding of our contributions due to unfamiliarity with this specific task domain. Should you require further or more detailed clarification, please feel free to reply. I am more than happy to provide additional explanation.

---

### Official Review · Reviewer_HTwH · 2025-11-03

**Soundness:** 1
**Presentation:** 1
**Contribution:** 1
**Rating:** 2
**Confidence:** 4

**Summary:**

This work proposes a video classification algorithm based on Fourier transform. FT is applied on both the visual frames and the audio. The coefficients are then filtered and fused for improved classification for the application of deception detection.

**Strengths:**

Deception detection is an interesting application.  Authors have shown improved results on three datasets.

**Weaknesses:**

1. A comparison of Tables 1, 2, 3 shows that performance of SPOT-JS is better with video modality compared to the fused modality.  It shows that the fusion method is not effective. Sections 4.5 and 4.6 has proposed the fusion method which actually reduced the performance.  Authors should avoid fusion and just re-write paper with only visual modality. As audio is not helpful, therefore any thing related to audio becomes redundant.

2. Different modules are presented as contributions.

3. Trivial things such as extracting frames from a video are very formally presented such as Temporal Deception Alignment Module (TDAM).

4. Some equations such as Eq. (16) looks arbitrary or AI generated.

**Questions:**

If fusion reduces performance then what is need to add fusion?

---

> ### Author Response · Authors · 2025-11-19
> **(1) Response to Reviewer HTwH weak1**
>
> Thank you for your valuable feedback. Please allow me to explain:
> (1) Regarding your first concern, while our method does achieve strong performance with the video modality, there are indeed cases where the fused modality outperforms video alone. Moreover, we have included ablation studies in the paper that demonstrate the effectiveness of our fusion approach. Taking the baselines on the RLT and DOLOS datasets as examples, although most fusion methods currently yield lower performance than video alone, this precisely highlights a critical research challenge: how to better integrate multiple modalities in deception detection. It is to address this very question that we proposed our fusion method.
>
> Deception detection is inherently complex. Identifying deceptive cues from the audio modality is particularly challenging, as features such as pitch variation, speech rate changes, and vocal tremors can all serve as potential indicators. However, extracting reliable signals from these features is difficult, which is why traditional psychologists often prefer visual cues—such as facial muscle movements, pupil dilation, and blink rate—as they are more readily observable.
>
> That said, the practical application of deception detection often involves ethical and logistical constraints. For instance, in criminal investigations, legal or situational factors may prevent the collection of video data, leaving audio as the only available modality. Therefore, audio-based deception detection remains highly valuable and is by no means redundant. Our work focuses on multimodal deception detection precisely because each modality can be critical in real-world scenarios where only one modality may be available.
>
> Currently, the field of multimodal deception detection lacks a mature and universally effective fusion framework. One of the key objectives of this paper is to shed light on the challenges in multimodal fusion—such as feature misalignment and noise interference—and to encourage the development of more effective fusion mechanisms. Abandoning fusion research due to its current suboptimal performance would hinder long-term progress in the field.
>
> It is also important to note that, given the difficulty of audio-based deception detection, improperly fused features can indeed mislead the model. This is one of the core challenges in this domain, especially as baseline accuracy increases and minor artifacts can significantly impact results. Nevertheless, the modules we proposed have been empirically shown to improve fusion performance, thereby contributing to the advancement of multimodal deception detection.
>
> In summary, both our fusion method and the inclusion of the audio modality are substantiated by experimental evidence and practical necessity. We believe they provide meaningful value to the research community and pave the way for more robust multimodal solutions in the future.

---

> > ### Comment · Reviewer_HTwH · 2025-11-23
> > **Usefulness of the proposed fusion algorithm**
> >
> > I have re-evaluated the effectiveness of the fusion algorithm using the ablation results presented in Table 4.  In the vast majority of experiments—over 90%—fusion actually leads to reduced performance. This suggests that, rather than providing complementary information, the fusion scheme may be introducing noise or irrelevant features. The authors should carefully revisit the fusion strategy to investigate why it is degrading performance.

---

> ### Author Response · Authors · 2025-11-19
> **(2) Response to Reviewer HTwH weak2**
>
> (2) Regarding your second question, each module we proposed has made distinct contributions, which is why we have listed them all as contributions in our paper. We have conducted corresponding experiments to empirically validate the effectiveness of these modules, and we believe this approach is fully justified.

---

> > ### Comment · Reviewer_HTwH · 2025-11-23
> > **How to state contributions**
> >
> > Contributions can be new algorithms or models, new theoretical results, new empirical conclusions. Instead of writing the names of the modules, authors should write what is new in their work.

---

> ### Author Response · Authors · 2025-11-19
> **(3) Response to Reviewer HTwH weak3**
>
> (3) Thank you for your question. Regarding the TDAM module, we acknowledge that we gave it a somewhat fancy name, and we appreciate your feedback. We welcome any naming suggestions you might have. However, the reason we formally named this module is because we believe it provides substantial practical value.
>
> First, this module enables automated frame sequence extraction and audio file retrieval directly from video files for deception detection tasks, eliminating the need for manual extraction of individual frames and audio files. Given the specific nature of our task, the datasets primarily contain video files without accompanying audio, which would typically require separate audio extraction—a cumbersome process that our module conveniently streamlines.
>
> From a technical perspective, our proposed TDAM differs significantly from previous data preprocessing methods. Traditional approaches typically process individually saved PNG images or handle frames independently, whereas our method processes continuous frame sequences. This approach offers substantial advantages for deception detection, as this task heavily relies on temporally dependent cues. As mentioned in our paper, traditional deception detection by psychologists involves observing muscle movements, pupil variations, head motions, and other time-dependent indicators that serve as crucial deception clues. Processing individual images or isolated frames would inevitably lose these vital temporal patterns, creating a significant limitation for deception detection. Unlike image classification tasks that can rely on single images, deception detection requires capturing sequential dependencies.
>
> To help illustrate this concept, consider the evolution from R-CNN to Fast R-CNN in object detection. Early R-CNN methods required extracting thousands of proposal regions from a single image, each requiring separate feature extraction and computation, resulting in significant computational overhead and redundant processing of overlapping regions. Fast R-CNN improved this by performing feature extraction on the entire image first, then generating bounding boxes—a simpler yet more effective approach. Similarly, our method represents a straightforward but effective solution for deception detection. We hope this explanation adequately addresses your concerns.

---

> ### Author Response · Authors · 2025-11-19
> **(4) Response to Reviewer HTwH weak4**
>
> (4) Regarding your fourth question, our formulas were rigorously derived and were not AI-generated. You mentioned that Formula (16) appears to be AI-generated, but in fact, it was designed based on inspiration from classical ResNet theory and guided by the Jensen-Shannon divergence formula.
>
> Let us first introduce some relevant background on JS divergence. Consider a travel scenario where we have two distributions, $P$ and $Q$. $P$ represents a perfect, precise, and real-time satellite map of a city, containing all true information such as streets, buildings, traffic conditions, and even temporary shortcuts. $Q$, on the other hand, is a simplified and possibly outdated hand-drawn map. Now, suppose you need to travel from point A to point B. With $P$, you can effortlessly plan the optimal route. However, when using $Q$, you might take wrong turns, follow longer paths, or even find that a shortcut on the map is actually a dead end. To correct your route based on these errors, you must expend extra time and effort. JS divergence can be understood as the "average extra cost" or "expected detour cost" you incur by relying on the imperfect map $Q$ instead of the perfect map $P$.
>
> If the imperfect map $Q$ is identical to the perfect map $P$, you incur no extra cost—the JS divergence is $0$. The more inaccurate $Q$ is, the more detours you must take, and the greater the JS divergence becomes. Now, let’s return to our Formula (16): the JS divergence $J$ measures the "distribution discrepancy cost" between the audio modality $Z^a$ and the visual modality $Z^v$. When $J$ is large, it indicates that the two modalities are like two highly inconsistent maps. Forcing fusion at this stage would result in a high "fusion cost," which would lead to poor fusion performance. In this case, the term $J Z^a + J Z^v$ in the formula retains more of the original information from each modality (akin to referring to the independent information from each map). When $J$ is small, it signifies high consistency between the two modalities. The term $(1-J)(W^a Z^a + W^v Z^v)$ then enables deep fusion (similar to integrating the advantageous information from two high-quality maps). Even when the two modalities are generally consistent ($J$ is small), $W^a$ and $W^v$ can still be optimized to learn the best weight combination for the specific task—just as, even with two accurate maps, we may still need to adjust the route planning strategy depending on the travel purpose (e.g., walking, driving, or sightseeing).
>
> We hope this explanation addresses your question. Thank you again for your inquiry.

---

### Comment · Area_Chair_XLg3 · 2025-11-25
**Authors' responses**

Dear Reviewers,

The authors have submitted their responses to your questions and feedbacks. Please read them and give your comments.

Regards,
AC

---

### Comment · Area_Chair_XLg3 · 2025-11-28
**The Author/Reviewer Discussion Phase deadline is approaching**

Dear Reviewers,

The Author/Reviewer Discussion Phase deadline is approaching. If you have not responded to authors’ rebuttal, please read it and give your feedback asap.

Regards,
AC

---

### Note · Authors · 2026-01-26

I have read and agree with the venue's withdrawal policy on behalf of myself and my co-authors.

---

### Meta-Review · Area_Chair_M3NS · 2026-01-04

**Summary:**

The main concerns raised by the reviewers include unclear presentation, excessive use of jargon, insufficient theoretical explanations of individual components, and doubts about the effectiveness and interpretability of the multimodal fusion design. Some reviewers questioned whether audio–visual fusion consistently outperforms models using only visual information. In addition, there were comments noting that the cited literature is outdated and that certain equations lack sufficient justification. Nevertheless, the reviewers also acknowledged the novelty of performing feature enhancement and fusion in the frequency domain, the principled use of Optimal Transport and JS divergence, and the extensive experiments demonstrating strong performance on cross-domain tasks.

**Reviewer Concerns:**

The reviewer raised concerns about insufficient theoretical explanations for each component and questioned the effectiveness and interpretability of the multimodal fusion design. The authors have addressed these issues through experiments. However, the issue regarding “A comparison of Tables 1, 2, 3 shows that performance of SPOT-JS is better with video modality compared to the fused modality” still needs a direct response. Specifically, when fusion is involved, the authors need to address why the fused modality underperforms compared to the single modality. Additionally, the readability of the paper is poor, and the authors should further improve the presentation. Overall, in response to the issues raised by reviewer HTwH, the authors should provide detailed and well-justified experimental evidence as well as clear mathematical derivations to support their claims. Thorough and rigorous experiments are necessary to convincingly explain the method and to ensure the soundness and credibility of the paper.

**Reviewer Scores:**

Reviewer 7UMw noted the lack of theoretical analysis and detailed discussion of the mechanisms, as well as issues with the paper’s readability. After the authors’ rebuttal, the reviewer is likely to increase their score. Reviewers HTwH and 18Hc are likely to maintain their current scores. Although the authors responded to the reviewers’ comments, their rebuttal lacks sufficient experimental evidence and theoretical support, making it unlikely to persuade the two reviewers to change their current evaluations.

---

### Decision · Program_Chairs · 2026-01-26

Reject